# Herbals and Plants in the Treatment of Pancreatic Cancer: A Systematic Review of Experimental and Clinical Studies

**DOI:** 10.3390/nu14030619

**Published:** 2022-01-30

**Authors:** John K. Triantafillidis, Eleni Triantafyllidi, Michail Sideris, Theodoros Pittaras, Apostolos E. Papalois

**Affiliations:** 1GI Department, Metropolitan General Hospital, 15562 Holargos, Greece; jktrian@gmail.com; 2Hellenic Society of Gastrointestinal Oncology, 354, Iera Odos Street, Haidari, 12461 Athens, Greece; eltriant@yahoo.gr; 3Women’s Health Research Unit, Queen Mary University of London, London E1 2AB, UK; mchsideris@gmail.com; 4Hematology Laboratory-Blood Bank, Aretaieion Hospital, School of Medicine, National and Kapodistrian University of Athens, 11528 Athens, Greece; teopittaras@yahoo.gr; 5Special Unit for Biomedical Research and Education, School of Medicine, Aristotle University of Thessaloniki, 60 El. Venizelou Street, Aghia Paraskevi, 15341 Athens, Greece

**Keywords:** alternative treatment, alternative medicine, chinese medicine, herbals, pancreatic cancer, medical plants

## Abstract

Background: Pancreatic cancer represents the most lethal malignancy among all digestive cancers. Despite the therapeutic advances achieved during recent years, the prognosis of this neoplasm remains disappointing. An enormous amount of experimental (mainly) and clinical research has recently emerged referring to the effectiveness of various plants administered either alone or in combination with chemotherapeutic agents. Apart from Asian countries, the use of these plants and herbals in the treatment of digestive cancer is also increasing in a number of Western countries as well. The aim of this study is to review the available literature regarding the efficacy of plants and herbals in pancreatic cancer. Methods: The authors have reviewed all the experimental and clinical studies published in Medline and Embase, up to June 2021. Results: More than 100 plants and herbals were thoroughly investigated. Favorable effects concerning the inhibition of cancer cell lines in the experimental studies and a favorable clinical outcome after combining various plants with established chemotherapeutic agents were observed. These herbals and plants exerted their activity against pancreatic cancer via a number of mechanisms. The number and severity of side-effects are generally of a mild degree. Conclusion: A quite high number of clinical and experimental studies confirmed the beneficial effect of many plants and herbals in pancreatic cancer. More large, double-blind clinical studies assessing these natural products, either alone or in combination with chemotherapeutic agents should be conducted.

## 1. Background

Pancreatic cancer (PC) represents the seventh most common cause of cancer-related mortality worldwide, with an average five-year survival rate of only 9%. According to GLOBOCAN 2018 estimates, PC caused 432,242 new deaths among 458,918 new cases that were reported during this year [1]. In the European Union, PC occupies the 4th position in cancer mortality [2]. It has been calculated that by 2040, the total number of cases in the European Union will increase by at least 30%. Even in Asia, the 5-year survival rate of PC is almost similar to that of the Western population: only 7.2% [3]. The exact etiology of this cancer remains unknown, although many factors have been proposed [4]. Genetic mutations, such as of *Kras* oncogene, inactivation of tumor-suppressor genes, telomere shortening, chromosomal loss, and gene amplification, have also been described and proposed. A meta-analysis revealed positive associations between PC risk and animal products, and starch-rich and Western dietary patterns. On the other hand, inverse associations between risk of PC and fruits, vegetables, vitamins (particularly vitamins D and B_12_ [5]), and fiber consumption [6] were noticed. Healthy drinking patterns may decrease the risk of PC, whereas heavy drinking may actually increase the risk [7]. Despite the fact that the management of PC has been remarkably improved, only 10% of PCs are resectable at the time of diagnosis [8].

The most important aspect concerning the treatment of PC is the poor response to the currently available chemotherapy. However, many plants and herbals have been used for centuries in the treatment of PC. Based on accumulating clinical and experimental data, it seems that the traditional Chinese medicine, as well as many plants growing in other parts of the world, should attract more attention by the scientific community concerning their exact role in the treatment of these patients. During the last decade, a real explosion in the number and quality of experimental and, to a lesser degree, clinical studies exploring the role of plants and herbals in PC was noticed [9].

The aim of this study was to collect and describe the existing experimental and clinical data concerning the therapeutic efficacy of plants and herbals in the human PC. We humbly anticipate that the content of this review could help the reader identify those plants and herbals having a scientifically proved antineoplastic action, and to easily separate the effective from the non-effective plant treatments. Another aim of this review was to force the health professionals, pharmaceutical companies, and health authorities to be aware of the necessity of performing clinical trials using plants, either alone or in combination with the conventional chemotherapy. We propose that medical schools should include at least some elementary data in their regular educational programs regarding the role of herbals and plants in various clinical disorders, including malignant neoplasms.

## 2. Methods

The study was conducted according to the PRISMA guidelines. We searched various international data bases, including PubMed, Web of Science, and Google Scholar, using the key words “plant”, “herbals”, “therapy”, “phytotherapy”, and “pancreatic cancer”. We also searched various other sources in order to obtain information concerning the physicochemical and other characteristics of the reported herbals and plants. All the in vivo and in vitro studies were identified and screened. Data were collected up to June 2021. The studies identified, either experimental or clinical, were categorized into five categories according to their subject and task: (i) clinical studies, e.g., studies exploring the role of certain plants in patients with PC; (ii) studies searching the role of herbals in PC stem cells; (iii) studies combining nanomedicine with herbals; (iv) studies investigating the role of combining plants with chemotherapy; and finally, (v) studies investigating the antiproliferative action of plants in PC cell lines, a category representing the great majority of the available studies. In order to help the reader see some of the details of one particular plant, the data were quoted in alphabetical order. An attempt was also made to include the chemical structure of the described plants and herbals, at least as a rough description, since the vast majority of plants and herbals consist of a large number of substances or groups of substances. In some of them, their chemical structure was described schematically. Flow information through the different phases of this systematic review is given in Figure 1.

A total number of 183 papers hes been excluded because of referring to reviews (*n* = 126), systematic reviews (*n* = 10), and meta-analyses (*n* = 8). Clinical trials (*n* = 26) and randomized clinical trials (*n* = 13) were also excluded, mainly because they were referring to dietary patterns, quality of life, and other parameters unrelated to the influence of herbals and plants on the clinical course of patients with PC.

## 3. Results

From January 1993 to June 2021, 125 experimental and clinical studies dealing with the influence of plants and herbals in PC were included. In the present systematic review, we were able to identify 86 studies looking at the effects of 74 different herbals and plant derivatives investigated in experimental models of PC cell lines and PC xenografts. We identified also a small number of clinical studies investigating the role of adding plants or plant extracts in the chemotherapy in patients with advanced PC. Regarding clinical studies exploring the effect of plant derivatives on PC stem cells, we identified five relevant studies. Concerning studies combining nanotechnology with herbals and plants aiming to target PC cells, we identified seven. Finally, we included a satisfactory number of articles referring to 22 different plants and herbals, looking at the cytotoxic effects of the co-administration of chemotherapeutic agents with plants. The majority of the plants studied grow in Asian countries, mostly in China and India, whereas the rest are plants growing in other parts of the world.

### 3.1. Clinical Studies Using Plants in the Treatment of Pancreatic Cancer

So far, very few clinical studies concerning the use of plants in the treatment of patients with PC have been conducted. However, almost all of them, including two case-reports, showed clear clinical benefit, meaning that more studies with a satisfactory number of patients should be performed. The plants used in these studies are analyzed below.

#### 3.1.1. PHY906

PHY906 consists of six herbs, namely *Scutellaria baicalensis Georgi, Glycyrrhiza uralensis Fisch., Ziziphus jujuba Mill.,* and *Paeonia lactiflora Pall* [10]. A large number of chemical substances, including 126 small molecules and 6 polysaccharides, have been isolated from *Scutellaria baicalensis*, mainly belonging to the categories of flavonoids and glycosides [11]. *Glycyrrhiza uralensis* encompases bioactive molecules, such as triterpene saponins and flavonoids [12]. Concerning *Ziziphus jujuba Mill, 22* compounds were identified in the ethanol extract. Four parent compounds and four metabolites were also detected in rat serum [13]. Finally, *Paeonia lactiflora* contains polyunsaturated and monounsaturated fatty acids, especially oleic and α-linolenic acids [14]. It seems that inhibition of CYP3A4; modulation of cytokines, macrophages, and lymphocytes; and inhibition of NF-kB, beta-glucuronidase, and the delta-opioid receptor represent the most significant mechanisms of action of PHY906. This mixture of herbs seems to be quite safe if it is consumed in doses less than 2.4 g per day. In a phase II study, 25 patients with advanced PC, previously treated with gemcitabine-based regimens, were treated with PHY906 plus capecitabine [15]. The results showed that the median progression-free survival and median overall survival were 10.1 weeks and 21.6 weeks, respectively. Eighteen patients, who received at least two cycles, achieved a median progression-free survival of 12.3 weeks and a median overall survival of 28 weeks. Six-month survival rate was 44%.

#### 3.1.2. *Viscum album* Extract

*Viscum album* is a hemi-parasitic shrub, growing on the stems of other trees, found in Europe and Western and Southern Asia. It contains several proteins, polysaccharides, phenolics, and various other compounds responsible for the chemical features of this product [16]. *Viscum album* extracts have cytotoxic, apoptotic, and immune stimulatory properties. Viscumin, the cytotoxic protein of the *Viscum album*, inhibits protein synthesis by binding to galactose residuals of the cell surface glycoproteins, and inactivating the 60S ribosomal subunit. Tröger et al., using *Viscum album extract* three times a week in escalating doses (0.01–10 mg) with no concurrently administed antineoplastic therapy in 220 patients with locally advanced or metastatic PC, showed that median overall survival was 4.8 months for *Viscum album* compared to 2.7 months for control patients. Within the “good” prognosis subgroup, median overall survival was 6.6 vs. 3.2 months, whereas within the “poor” prognosis subgroup, it was 3.4 vs. 2.0 months, respectively. No adverse events were observed. In this trial, *Viscum album* therapy resulted in a prolongation of the overall survival rate [17]. Werthmann et al. also described a case of a patient with advanced PC with R1-resection who developed liver metastasis. The patient received concomitant FOLFIRINOX, and, after that, *Viscum album* was added. With this regime, a patient survival and patient relapse-free survival of 63 and 39 months, respectively, was achieved [18].

#### 3.1.3. Mistletoe

Mistletoe is an obligate hemiparasitic plant having, as main constituents, iscotoxins, lectins, flavonoids, sterols, and alkaloids [19]. Phoratoxin and tyramine are responsible for some side-effects, including diarrhea, vomiting, cardiac rate disturbances, and cardiac arrest. It has been used among other treatments as a complementary treatment for cancer patients. However, the use of this plant for the treatment of cancer has not been approved by the FDA of the USA.

Mistletoe leaves are quite popular in Europe for their immunomodulatory and cytotoxic properties, including the induction of apoptosis, and the trapping of chemotherapeutic drugs within cancer cells through the inhibitory potential of P-glycoprotein. Regarding clinical data, Matthes et al. conducted an observational pharmaco-epidemiological study in patients operated on for PC. Patients were treated with gemcitabine supported by mistletoe, or with gemcitabine alone. Mistletoe improved the symptoms and overall survival of the patients. Therefore, it could be used as supportive care within gemcitabine protocols in patients operated on for PC [20]. A case report also stressed the clinical benefits derived from the administration of this plant. Ritter et al. described the case of a woman aged 43 who had undergone a pancreatic head resection plus nine cycles of palliative treatment with gemcitabine plus oxaliplatin, and concomitant treatment with mistletoe. Ten months later, the patient was free of any evidence of tumor progression [21].

### 3.2. Effect of Plant Derivatives on PC Stem Cells

A large number of cancer stem cells have been identified in PC tissues. These cancer stem cells seem to play a significant role in the tumor initiation and evolution, and also have the ability to divide them and differentiate exactly as the parent stem cell. Another quite significant characteristic of these cells is their ability to be resistant to the current chemotherapeutic agents. It is obvious, therefore, that any one anticancer therapy, in order to be successful, should also target cancer stem cells [22]. The relevant studies are analyzed subsequently.

#### 3.2.1. Bitter Melon Juice

The main constituents of bitter melon are triterpene, proteid, steroid, alkaloid, inorganic, lipid, and phenolic compounds. Several glycosides have also been isolated from the *Momordica charantia* [23]. Dhar et al. [24] investigated the efficacy of bitter melon juice against CD44^+^ /CD24^+^ /EpCAM^high^ enriched PC stem cells in spheroid assays. They found that bitter melon juice increased the sensistivity of gemcitabine-resistant PC stem cells, and decreased the expression of genes involved in stem cell proliferation. However, further investigation of its efficacy against PC, including gemcitabine-resistant cases, is urgently needed.

#### 3.2.2. Crocetinic Acid (*Gardenia jasminoides*)

A number of chemical components of *Gardenia jasminoides* have been isolated and characterized, including iridoids, glucosides, and triterpenoids. Three major bioactive compounds have been isolated, namely geniposide, genipin, and gardenoside [25]. Moreover, *Gardenia jasminoides* contains crocetin, a natural apocarotenoid dicarboxylic acid. This chemical substance represents a carotenoid constituent of saffron that is showning anti-tumor action in animal models. Rangarajan et al. found that crocetinic acid inhibited the proliferation of PC cell lines, and induced apoptosis in a dose- and time-dependent manner. Crocetinic acid also inhibited the epidermal growth factor receptor, decreased the number and size of the pancospheres, and suppressed the expression of the marker protein doublecortin calcium/calmodulin-dependent kinase-1. Crocetinic acid, at a dose of 0.5 mg/Kg BW, also suppressed the growth of tumor xenografts [26].

#### 3.2.3. Hispidin (*Phellinus linteus*)

Lee et al. purified six compounds from the mushroom *Phellinus linteus* using reversed-phase high-performance liquid chromatography, and also identified their structures, by spectroscopic methods, as caffeic acid, inotilone, 4-(3,4-dihydroxyphenyl)-3-buten-2-one, phellilane H, (2*E*,4*E*)-(+)-4′-hydroxy-γ-ionylideneacetic acid, and (2*E*,4*E*)-γ-ionylideneacetic acid; the first three exhibiting the most potent antioxidant activity [27]. This mushroom contains also hispidin, a polyphenolic compound that has been shown to have antineoplastic properties. Chandimali et al. evaluated the cytotoxic effects of hispidin on BxPC-3 and AsPC-1 PC cells. They also investigated the possible synergistic effect of gemcitabine and hispidin on PC stem cells in vitro. They found that hispidin exerted antitumor effects against both PC cells and stem cells. Furthermore, hispidin sensitized PC stem cells to gemcitabine, thus promoting its therapeutic efficacy [28].

#### 3.2.4. Quercetin and Resveratrol

Quercetin is a flavonol comprising fifteen carbon atoms, with two aromatic rings connected by a three-carbon bridge. The main structure of benzopyran-4-one makes it a hydrophobic compound. The rich food sources of quercetin are onions, berries, and apples [29]. Resveratrol is a compound that belongs to the category of phytoalexins (based on its function), and to the category of stilbenoids (based on its structure), since its basic structure is stilbene [30]. In a study aiming to investigate the effect of these phytochemicals on epithelial-mesenchymal transition of PC stem cells, Hoca et al. submitted CD133+ and CD133− PANC-1 cells on different concentrations of resveratrol and quercetin. They found that the immunostaining intensity of CD133+ cells was stronger than CD133− cells. ACTA-2, IL-1β, and N-cadherin immunoreactivities were decreased, whereas TNF-α and vimentin immunoreactivities increased in quercetin-treated CD133+ cells [31].

#### 3.2.5. Pao Pereira

Pao pereira is an indigenous tree growing in the Amazon rainforest. The active components, mainly indole alkaloids and beta-carboline alkaloids, were derived from extracts of plants of the same genus [32]. It has been used as an antipyretic, antiviral, antimalarial, and anticancer agent, either alone or in combination with other chemotherapeutic agents. In a relevant study, it was found that pao pereira inhibited proliferation of PC cell lines, and reduced the population of PC stem cell lines. Moreover, Nuclear β-catenin levels were decreased, suggesting suppression of the Wnt/β-catenin signaling pathway. Finally, in in vivo experiments, pao pereira reduced the tumorgenicity of PC cell lines in immunocompromised mice [33]. Pao Pereira represents a novel therapeutic strategy targeting PC stem cells.

### 3.3. Nanotechnology to Target PC

The role of nanotechnology in cancer treatment is to formulate and improve the physicochemical properties of anticancer drugs (e.g., aqueous solubility or circulation time after administration) in order to increase their efficacy. So far, a small number of studies have been published regarding the use of nanotechnologies in patients with PC in combination with gemcitabine and other molecules. Among the nanotechnologies used in PC, some are site-specific, whereas others have been used as image guidance and controlled release. Nanotechnology poses a number of advantages compared to traditional technologies because it can guide drug molecules to the target destination. On the other hand, the size of nanoparticles (1–100 nm) improves the diagnostic and therapeutic confrontation of PC via interaction with biological molecules [34]. Selective release of anticancer drugs into the cancer tissues is extremely difficult, although nanotechnology comes to overcome this difficulty. Due to their large size, intravenously administered nano-sized macromolecular chemotherapeutic drugs could not be excreted into the urine neither penetrate the tight junctions of endothelial cells of the normal blood vessels thus increasing their plasma concentration and half-life time. On the contrary, they can penetrate the malignant tissue due to the existence of an abundance of vascular abnormalities (hypervascularization, aberrant vascular architecture, etc.) present in the neoplastic tissue favoring this penetration [35,36]. Therefore, high concentrations of the chemotherapeutic drug in the cancerous tissues could be achieved, resulting in improvement of the therapeutic effect and reduction of the rate of side effects. Finally, binding nanoparticles with chemotherapeutic agents can enhance tumor penetration, increase the circulation time of the drug, reduce the rate of liver and kidney detoxication, and prevent the toxic effects of the use of adjuvants [37]. Although a large number of nanoparticle-drug-combined formulations exist [38], there is a limited number of experimental studies combining nanotechnology with herbals and plants aiming to target PC. These studies are analyzed subsequently.

#### 3.3.1. Anthothecol-Encapsulated PLGA Nanoparticles (Khaya Anthotheca—Meliaceae)

Anthothecol, an effective antimalarial compound, is a limonoid isolated from the plant Khaya anthotheca, the chemical structure of which is shown in Figure 2.

In a relevant study, Verma et al. found that anthothecol-encapsulated PLGA-nanoparticles induced apoptosis in PC stem cells and cancer cell lines, and reduced cell motility, migration, and invasion [39].

#### 3.3.2. *Eysenhardtia platycarpa*

*Eysenhardtia platycarpa* is a plant whose extracts have been used in Mexican traditional herbal medicine. The health-related beneficial properties of the plant are related to the contained phenolic compounds, mainly flavonoids, a group of benzo-pyrone derivatives with potential anti-inflammatory properties [40]. *Eysenhardtia platycarpa* leaves contain the flavonone (2*S*)-5,7-dihydroxy-6-prenylflavanone. This compound has been used in order to generate cytotoxic derivates. The compounds 1, 1a, 1b, 1c, and 1d, and compounts encapsulated in polymeric nonoparticles (NPs1, NPs1a, NPs1b, NPs1c, and Nps1d), showed cytotoxic activity against the PC cell line [41]. The NPs1a nanoparticle could be considered as a promising agent against PC.

#### 3.3.3. *Panax notoginseng* Gold Nanoparticles

*Panax notoginseng,* a species of the genus *Panax,* is commonly referred as Chinese ginseng because it grows naturally in China. The major constituents of *Panax notoginseng* are dammarane-type ginsenosides [42]. In traditional Chinese medicine, the regulal dose of *Panax notoginseng* decoction is 5–10 g. Wang et al. [43] synthesized gold nanoparticles from Panax notoginseng. These gold nanoparticles induced cytotoxicity and apoptosis in PC cell lines via inceased apoptotic gene expression.

#### 3.3.4. Parvifloron-D-Loaded Smart Nanoparticles

*Plectranthus* is a plant growing mainly in the Southern hemisphere. It includes more than 350 species, with some of them used as ornamental plants, leaf vegetables, and root vegetables. Isolated compounds of the *Plectranthus* genus, such as parvifloron D (Derris parviflora) growing in Sri Lanka, were found to have cytotoxic and antiproliferative activities. Santos-Rebelo et al. [44] evaluated the antiproliferative effect of parvifloron D isolated from *P. ecklonii* against PC cell lines using albumin nanoparticles. They found that parvifloron D has selective and significant cytotoxicity against PC lines.

#### 3.3.5. *Silibinin*

*Silymarin*, the extract from the medicinal plant Silybum marianum (milk thistle), consists primarily of silybin and its isomers, silicristin, and silidianin. This substance displays significant antioxidant- and membrane-stabilizing activity, thus protecting various organs against chemical injury. It can also enhance the regenerative ability of the liver [45]. Finally, these flavonoid agents have also been shown to exert significant anti-neoplastic effects in a variety of in vitro and in vivo cancer models. In a relevant study [46], it was shown that Silibinin encapsulated in polymersome nanoparticles induced apoptosis, and inhibited migration and proliferation in PC cells and cancer stem cells, through the suppression of onco-miRNAs, and the induction of tumor suppressive miRNAs.

#### 3.3.6. Zinc Oxide Nanoparticles Using *Anacardium occidentale* Leaf Extract

The tropical tree *Anacardium occidentale* produces the cashew seed and apple. The chemical components consist mainly of glutelin, albumin, globulin, and other protein isolates [47]. It should be stressed that cashews can cause severe allergic reactions in a proportion higher than 6%. Zhao et al. synthesized zinc oxide nanoparticles with *Anacardium occidentale* leaf extract by boiling the mixture of 10 mL of *Anacardium occidentale* leaf extract and 30 mL 0.1 M zinc nitrate at 60 °C for 3 h. The results revealed that that the zinc oxide nanoparticles exhibited significant cytotoxicity against PC cell lines in a dose-dependent manner [48].

#### 3.3.7. *Scutellaria barbata* Gold Nanoparticles

*Scutellaria barbata* is a perennial herb prevalent in Korea and southern China. Concerning the chemical compounds of this herb, it seems to contain flavonoids, diterpenoids, polysaccharide, volatile oil, and steroids [49]. This herb has been used in China and Korea in patients with malignant disorders [50]. In addition to anticancer properties, *Scutellaria barbata* has been reported to have anti-inflammatory, antioxidant, and antimicrobial properties. Wang et al. synthesized gold nanoparticles from the *Scutellaria barbata* using a green route method, and evaluated its anticancer activity against PC cell lines. The synthesized gold nanoparticles showed strong anticancer activity [51]. Further research on this plant could help the development of novel anticancer drugs against PC.

### 3.4. Combined Administration of Plants with Chemotherapeutic Agents in Patients with PC

The nucleoside analogue gemcitabine is considered to be one of the most important chemotherapeutic agents for the treatment of PC. However, it shows a low response rate and free survival due to the development of chemoresistance. In clinical practice up to 50% of patients with gemcitabine-pretreated disease are submitted to further treatment. The median survival rate, with the best supportive care, in patients who have failed gemcitabine is no longer than two months. Therefore, the need for new agents for advanced PC in cases of gemcitabine resistance is of great importance for the patients’ outcome. During the last years, a number of plants and plant derivatives administered in combination with gemcitabine have shown promising anticancer results by targeting many signaling pathways in in vitro and in vivo PC models [52]. These plants (in alphabetical order) are shown in Table 1, and summarized subsequently.

#### 3.4.1. Asparagus

Asparagus extract is a natural remedy sourced from the spears, root, and rhizomes of the asparagus plant, used in alternative and ayurvedic medicine for urinary disorders. The major bioactive constituents of asparagus are steroidal saponins. Other chemical constituents of Asparagus are essential oils, asparagine, arginine, tyrosine, flavonoids resin, and tannin [77]. It is likely safe when eaten in recommended amounts. A product containing asparagus root and parsley leaf is unsafe when taken at doses more than 6 g/d. There are no guidelines for the appropriate use of asparagus extract, although dosages of up to 150 mg/d have been used in short-term studies with no reported side-effects. Shimada et al. [53] showed that enzyme-treated asparagus extract down-regulated heat shock protein27 in klm1-r cells, suggesting their potential therapeutic benefit in enhancing anticancer effects by its combination with gemcitabine.

#### 3.4.2. Chokeberry Extracts (*Aronia melanocarpa*)

*Aronia melanocarpa*, the black chokeberry, is a species of shrub in the rose family native to eastern North America. Polyphenols are biofactors that determine the high bioactivity of chokeberries, and include proanthocyanidins, flavonols, flavanols, proanthocyanidins, and phenolic acids [78]. Most of the favorable effects of *Aronia melanocarpa* anthocyanins are due to their high antioxidative activity. They are hepatoprotective with a significant anti-inflammatory and bacteriostatic activity in vitro against microbes and viruses. Finally, they have antimutagenic activity, and suppress the growth of human colon cancer cells. There are no reports regarding the side-effects of this plant. Thani et al. investigated the pro-apoptotic effects of chokeberry extract in a human PC cell line, and it possibly enhanced cytotoxicity in combination with gemcitabine. It was found that the combination was more effective than gemcitabine alone [54].

#### 3.4.3. *Coix* Seed Emulsion

The pharmacological extract of *Coix* lachrymal-jobi seeds, a cereal grain mainly popular in tropical Asia, represents the most commonly used anticancer agent in China. An NFkB-depended assay demonstrated dose-dependent inhibition of NFkB signaling after treatment of cultures with this extract, associated with a reduced translocation of the Rel-A/p65 subunit of NFkB to the nucleus. *Coix* extract also inactivated protein kinase C, a major activator of NFkB [79]. Qian et al. showed that pre-treatment with *coix* seed emulsion synergistically sensitized PC cell lines to gemcitabine. Pre-treatment with c*oix* seed emulsion resulted in induction of proapoptosis proteins after lower doses of gemcitabine compared to monotherapy. Furthemore, *coix* seed emulsion suppressed the gemcitabine-induced activation of NF-kB, and down-regulated the anti-apoptotic molecules Bcl-2, surviving, and COX-2. In in vivo experiments, *coix* seed emulsion combined with gemcitabine had a higher antitumor activity compared to either agent alone [55].

#### 3.4.4. C5E

Pak et al. produced a novel herbal mixture extract cocktail containing 10 types of traditional Chinese medicine herbs (C5E) [56]. These authors investigated the anticancer effect of this herbal mixture in the PC cell line, PANC-1, in the absence or presence of gemcitabine. They found that the percentage of side population cells, and the cell viability of PANC-1 cells, were decreased in response to all treatments via induction of apoptosis. The mRNA expression levels of sonic hedgehog were down-regulated to a greater extent following the co-treatment with C5E and gemcitabine compared with the treatment with either C5E or gemcitabine alone, suggesting that the combined treatment may exhibit synergistic effects in PANC-1 cells.

#### 3.4.5. *Emodin*

*Emodin* is a trihydroxyanthraquinone that is 9,10-anthraquinone which is substituted by hydroxy groups at positions 1, 3, and 8, and by a methyl group at position 6. It is present in the roots and barks of numerous plants, moulds, and lichens. It derives from an emodin anthrone. The chemical structure of Emodin is shown in Figure 3. 

Wei et al. found that *emodin* suppressed tumor growth in mice inoculated with PC cells. The combination treatment with gemcitabine promoted apoptotic cell death and mitochondrial fragmentation, and reduced phosphorylated-Akt level, NF-κB activation, and Bcl-2/Bax ratio [57].

#### 3.4.6. *Escin*

*Aesculus hippocastanum* represents a well-known plant in Chinese medicine having anti-inflammatory, antianalgesic, and antipyretic activities. It is commonly known as a horse-chestnut or conker tree. *Escin* is the main active component in horse chestnut, and is responsible for most of its medicinal properties, although the mixture also contains various other components [80]. *Escin* appears to act through a wide range of mechanisms, including the induction of endothelial nitric oxide synthesis, and the release of prostaglandin F_2α_, serotonin, and histamine antagonism, as well as the reduction of the catabolism of tissue mucopoly-saccharides. Rimmon et al. showed that *Escin* decreased the survival of PC cells, and down-regulated the NF-κB signaling pathway. *Escin* combined with gemcitabine showed only an additive effect, whereas its combination with cisplatin resulted in a significant synergistic cytotoxic effect in PANC-1 cells [58].

#### 3.4.7. Fisetin

Fisetin is a 7-hydroxyflavonol with additional hydroxy groups at positions 3, 3′, and 4′ (3,3^′^,4^′^,7-OH flavone). It represents one of the most prevalent plant flavonoids present in many fruits and vegetables. It can modulate a number of cancer signaling pathways and growth factors, such as Akt, JNK, p38MAPK, NF-κB, and VEGF cytokines and chemokines, thus inducing apoptosis and cell cycle arrest [81]. The chemical structure of Fisetin is shoun in Figure 4.

Kim et al. showed that combination treatment with *fisetin* and gemcitabine inhibited the proliferation of PC cells, and induced apoptosis. They also showed that *Fisetin* sensitized human PC cells to gemcitabine-induced cytotoxicity through the inhibition of ERK-MYC signaling [59].

#### 3.4.8. *Gloriosa superba* L. (Glory Lily, Colchicaceae)

*Gloriosa superba* is a species of flowering plant in the family Colchicaceae. This plant contains colchicine, and alkaloids related to colchicines, such as 3-O-demethylcolchicine, have been used succesfully in the treatment of gout [82]. The plant is poisonous and toxic enough to cause death if ingested in large quantities. Every part of the plant is poisonous, especially the tuberous rhizomes. In a murine model of PC, Capistrano et al. evaluated a crude ethanolic extract and colchicine-poor/colchicoside-rich extract [60]. They were administered at a dose of 4.5 mg/kg (p.o., daily) total content of colchicine and derivatives during 3 weeks, or at a dose of 3.0 mg/kg (p.o., daily) combined with gemcitabine (60 mg/kg, i.p., 3×/week) for 54 days. The results revealed a significant delay in tumour growth over time for gemcitabine and the combination therapy compared to the control group. A significant prolongation of the survival of the groups treated with gemcitabine and the combination therapy was also observed.

#### 3.4.9. Herbal Mixture Extract

Herbal mixture extract is composed of three oriental herbal plants, 40% *Meliae fructus* (China), 40% *Cinnamon bark* (Vietnam), and 20% *Sparganium rhizome* (China), that are considered to have anticancer activity. In a relevant study, Pak et al. [61] showed that herbal mixture extract inhibited PC cell growth by promoting G0/G1 arrest and apoptotic cell death. It also suppressed stem-cell-like side population cells and migration activity. In a PC xenograft model, herbal mixture extract suppressed tumor growth by 46%, compared to a 36% decrease caused by gemcitabine. However, contrary to the in vitro results, combined treatment of herbal mixture extract with gemcitabine enhanced tumor growth, suggesting that this co-treatment is not beneficial for PC.

#### 3.4.10. *Isodon* Eriocalyx and Its Bioactive Component Eriocalyxin-b

*Isodon* is a group of flowering plants in the family lamiaceae. The plant is native to tropical and subtropical parts of the world. Li et al. showed that eriocalyxin-b, a diterpenoid isolated from isodon eriocalyx, possesses anti-PC effects. They also noticed that gemcitabine and eriocalyxin-b had a synergistic anti-proliferative effect, as both cellular apoptotic and anti-proliferative effects of gemcitabine were increased after combined administration with eriocalyxin-b. The mechanisms involved included increased activation of the caspase cascade, and induction of JNK phosphorylation. Therefore, gemcitabine and eriocalyxin-b taken together regulated pdk1/akt1/caspase and JNK signaling, and promoted apoptosis synergistically [62].

#### 3.4.11. Monogalactosyl Diacylglycerol

Monogalactosyl diacylglycerol, derived from spinach, possesses cytotoxic effects in human cancer cell lines. Akasaka et al. isolated monogalactosyl diacylglycerol from spinach. In separate experiments, they showed that gemcitabine and monogalactosyl diacylglycerol suppressed growth of PC cell lines via selective inhibition of replicative polymerase inhibitors species. Moreover, gemcitabine combined with monogalactosyl diacylglycerol had synergistic effects on the inhibition of DNA replicative polymerase inhibitors, compared with gemcitabine or monogalactosyl diacylglycerol alone [63]. In a subsequent study, the same group of researchers investigated the role of this compound in enhancing the effect of radiation on human PC cell lines and normal human dermal fibroblasts in vitro and in vivo. They noticed that monogalactosyl diacylglycerol showed a dose- and time-dependent cytotoxicity, as well as a reduction in the number of cell colonies, upon treatment with both monogalactosyl diacylglycerol and radiation as compared to irradiation alone. The combined radiation and monogalactosyl diacylglycerol treatment showed a higher proportion of apoptosis and DNA damage in malignant pancreatic cells, as compared to either one alone [64].

#### 3.4.12. *Moringa oleifera*

*Moringa oleifera* is a fast-growing tree native to tropical and subtropical regions of South Asia. It is widely cultivated for its young seed, pods, roots, flowers, and leaves, used as vegetables and for traditional herbal medicine. It has antiproliferative and antimetastatic properties. Some studies suggest that it may cause adverse effects when consumed in large quantities. The supplementation with *M. oleifera* leaf extract is potentially toxic at levels exceeding 3000 mg/kg of BW, being safe at levels below 1000 mg/kg. It may interfere with prescription drugs affecting cytochrome P450. Hagoel et al. have shown that *moringa* administration combined with radiation therapy significantly inhibited human PC cell survival, induced apoptosis, and reduced the metastatic activity of these cells. Combined treatment also resulted in a decreased expression of Bcl-2, and down-regulation of the PARP-1 and the NF-κB-related proteins. Moringa also inhibited the growth of tumors generated by human PC cells in nude mice. The combination of moringa with radiation exhibits an additional inhibitory effect by overcoming the radioresistance of PC cells [65].

#### 3.4.13. Nexrutine^®^

Nexrutine is an extract consisting of a blend of several active protoberberine alkaloids derived from the bark of *Phellodendron amurense* [83]. It has been utilized in traditional Chinese medicine as a potent antidiarrheal and antiinflammatory agent. Gong et al., using multiple human PC cells, found that the combination treatment with nexrutine and gemcitabine resulted in significant alterations of proteins in the STAT3/NF-κB signaling axis, and growth inhibition in a synergistic manner [66]. 

#### 3.4.14. Ocoxin Oral Solution

Ocoxin oral solution comprises a mixture of several natural compounds, such as green tea extract; glycyrrhizic acid; vitamin C, B_6_, and B_12_; minerals; and amino acids. This nutritional supplement possesses immunomodulatory, anti-inflammatory, and antioxidant properties, as well as antitumor effects, either alone or in combination with irinotecan in liver metastases from colorectal cancer [84]. Hernandez-Unzueta et al. investigated the effect of ocoxin oral solution in an experimental PC model, and its influence in stroma-related chemoresistance to paclitaxel and gemcitabine. They showed that this solution enhances the cytotoxic effect of paclitaxel and gemcitabine, and ameliorates the chemoresistance in human PC cells. It also promoted the expression of the altered genes, and decreased pancreatic tumor development in vivo [67].

#### 3.4.15. *Oplopanax horridus*

*Oplopanax horridus* (devil’s club or devil’s walking stick) is a large understory shrub, endemic to the rainforests of the Pacific Northwest. The main chemical constituents of *O. horridus* are polyynes phenylpropanoids lignan glycosides, triterpenoids, sesquiterpenes, and volatile compounds [85]. The plant is used in a variety of ways, most commonly as an oral tea in traditional settings. Native American populations have used the plant as traditional medicine for conditions such as diabetes and rheumatoid arthritis. Its root and stem bark extract showed antiproliferation activity. Cheung et al. [68] investigated the effects of devil’s club ethanol extract alone or in combination with cisplatin, gemcitabine, and paclitaxel on pancreatic endocrine HP62, and pancreatic ductal carcinoma PANC-1 and BxPC-3 cells. They found that devil’s club extract inhibited the proliferation of HP62, PANC-1, and BxPC-3 cells. Devil’s club combined with paclitaxel inhibited synergy on PANC-1 cells. An up-regulation of cytochrome C, claspin, cIAP-2, and HTRA2/Omi apoptosis-related markers in devil’s-club-treated HP62 and PANC-1 was also found. The extract acts through targeting the mitochondrial apoptosis pathway in the PC cells. In another study, Tai et al. [69] found that PANC-1 3D spheroids were more resistant to killing by *Oplopanax horridus* extract, gemcitabine, and paclitaxel compared to 2D cells. *Oplopanax horridus* extract enhanced the antiproliferation activity of cisplatin and gemcitabine. The use of a 3D spheroid model for the screening of natural products can increase the efficiency in discovering in vivo bioactive compounds.

#### 3.4.16. *Paeonia suffruticosa*

*Paeonia suffruticosa* has been used extensively in Chinese medicine. There are more than 1000 Chinese tree peony cultivars that have been selected for more than 2000 years. The root bark of *Paeonia suffruticosa* could inhibit the growth and metastasis of cancer, although the exact mechanism(s) of this inhibion are unknown. Liu et al. showed that the oral administration of *Paeonia suffruticosa* aqueous extracts, alone or in combination with gemcitabine, delayed tumor growth in a xenograft model by stimulating the endoplasmic-reticulum-related proteostasis stress, and inducing autophagy and cell apoptosis. This proteostasis impairment resulted in altered dynamics of the actin cytoskeleton, and cell cycle progression inhibition. It seems that reactive oxygen species generated by *Paeonia suffruticosa* may trigger mitophagy and, finally, cell apoptosis [70].

#### 3.4.17. *Pao Pereira*

*Pao Pereira* provides monoterpenoid indole alkaloid rich extracts and fractions used in clinical practice for the treatment of prostate cancer and AIDS. The extract of Pao Pereira, either alone or in combination with gemcitabine, induced dose-dependent apoptosis in all five tested PC cell lines. The combination with gemcitabine had a synergistic effect in the inhibition of cell growth. In an orthotopic pancreatic xenograft mouse model, *Pao Pereira* significantly suppressed tumor growth. Combined *Pao Pereira* and gemcitabine treatment further enhanced the tumor inhibitory effect compared to gemcitabine alone [71].

#### 3.4.18. Piperlongumine

Piperlongumine, a biologically active alkaloid/amide phytochemical extracted from long pepper, has several biological activities, including selective cytotoxicity against multiple cancer cells of different origins at a preclinical level. Rawat et al. showed that piperlongumine inhibits cell proliferation of INT-407 and HCT-116 cells, and increases the levels of intracellular reactive oxygen species. Moreover, P53, P21, BAX, and SMAD4 were up-regulated, whereas BCL2 and survivin were down-regulated [72]. The combination study confirmed the synergistic effect of piperlongumine with the chemotherapeutic agent paclitaxel.

#### 3.4.19. Resveratrol

This natural polyphenol is a phytoalexin exhibiting very high antioxidant and antimicrobial potential, acting against many pathogens, including bacteria and fungi. Other effects, such as cardioprotective, phytoestrogenic, and neuroprotective, have also been reported. Moreover, it could inhibit the growth of many cancers both in vitro and in vivo. At doses less than 1 g/d, resveratrol does not appear to cause side-effects. At doses of 2.5 g/d or more, side-effects, such as nausea, vomiting, diarrhea, and liver dysfunction, could be observed. However, in long-term clinical trials, no significant side-effects were noticed. In fact, resveratrol has been found to be safe and well-tolerated at doses up to 5 g/day [86]. In an experimental study, Jiang et al. found that resveratrol suppressed the proliferation, and induced apoptosis in PC cells via the activation of AMP protein kinase. They also noticed that the silencing of the YES-activated protein by resveratrol enhanced the sensitivity of PC cells to gemcitabine [73]. 

#### 3.4.20. *Rauwolfia vomitoria*

*Rauwolfia vomitoria* is a plant species of the genus *Rauvolfia*. It is native in countries of Africa, China, Bangladesh, and Puerto Rico. The plant contains a number of compounds widely used in traditional medicine [87]. An extract of the roots is extensively used in patients with diarrhea, jaundice, and abdominal colic or fever, and also to reduce blood pressure. The plant contains a number of compounds with pharmaceutical action, including reserpine, reserpinine, deserpidine, ajmalicine, and ajmaline. *Rauwolfia vomitoria* contains 2,6-Dimethoxybenzoquinone, which is a benzoquinone with antiproliferative action against malignant cells. However, many parts of the tree are toxic and should be used with caution. Yu et al. showed that *Rauwolfia vomitoria* induced apoptosis in a concentration-dependent manner. The combined administration of *Rauwolfia vomitoria* and gemcitabine had a synergistic effect in inhibiting cell growth. *Rauwolfia vomitoria* also suppressed tumor growth and metastatic potential in an orthotopic PC mouse model [74].

#### 3.4.21. Thymoquinone

*Nigella sativa* L. represents a source of a number of bioactive compounds, including thymoquinone, α-pinene, p-cymene, and monoterpenes [88]. Thymoquinone represents the predominant bioactive ingredient of Nigella sativa. This chemical has been shown to have anti-cancer and chemo-sensitizing effects on PC. The potency of the combined administration of thymoquinone and gemcitabine in inducing apoptosis and preventing gemcitabine-insensitivity in PC cells was recently investigated. It was found that thymoquinone pre-treatment following gemcitabine treatment increased the PC cell apoptosis, and inhibited tumor growth both in vitro and in vivo, by affecting multiple molecular signaling targets. The combination also induced down-regulation of anti-apoptotic, and up-regulation and activation of pro-apoptotic molecules [75]. Therefore, thymoquinone pretreatment can enhance the anti-cancer activity of gemcitabine.

#### 3.4.22. Triptolide

Triptolide is an organic heteroheptacyclic compound, an epoxide, a gamma-lactam, and a diterpenoid. The chemical structure of Triptolide is shown in Figure 5. 

Yang et al. analyzed the combined cytotoxic effect of triptolide and hydroxycamptothecin on the PC cell line PANC-1. They showed that the cytotoxic result of a combined therapy was superior to that of triptolide or hydroxycamptothecin alone. They also suggested that the activation of caspase-9/caspase-3, and inhibition of the NF-κB signaling pathway were the mechanisms responsible for the synergistic cytotoxic effect of this combination therapy [76]. Combined triptolide and hydroxycamptothecin therapy in patients with PC should be tested.

### 3.5. Experimental Studies Using Plants as Unique Agents against Pancreatic Cancer Cell Lines and Xenografts

The volume of experimental studies published so far includes 68 products of herbals or plants. Almost all experimental studies showed beneficial effects, strongly suggesting that these plants or plant derivatives should be tested in large clinical trials. In order to facilitate the reader to easily find out the details concerning a certain plant or plant extract, the names and mode of action of the natural products and herbals described in this review are shown in Table 2 (in alphabetical order), and analyzed subsequently.

#### 3.5.1. *Achyranthes Aspera*

*Achyranthes* aspera is a plant used as an anti-cancer agent in traditional medicine in India. Chemical investigation of the seeds of this plant resulted in isolation and identification of saponin A (as D-glucuronic acid) and saponin B (as β-D-galactopyranosyl ester of D-glucuronic acid). Certain other constituents were also isolated, including oleanolic acid, amino acids, and hentriacontane. The seeds also contain 10-tricosanone, 10-octacosanone, and 4-tritriacontanone [178]. The plant has been shown to exhibit time- and dose-dependent cytotoxicity on PC cells. It also selectively suppresses the transcription of metalloproteases, inhibitors of MMPs, and angiogenic factors [89].

#### 3.5.2. *Alpinia officinarum*

*Alpinia officinarum* is a plant in the ginger family, cultivated in Southeast Asia. The rhizomes of the plant have been used in curries and perfumes. It contains high amounts of the flavonol galangin. Dong et al. showed that these diarylheptanoids contained in *Alpinia officinarum* suppressed cell proliferation, and induced the cell cycle arrest of PC cells [90].

#### 3.5.3. *Amoora rohituka*

*Amoora* rohituka is a plant found in many districts of Bangladesh. The species of the plant contain several triterpenoids, including limonoids, steroids, an alkaloid, a chromone (noreugenin), three flavonoid glycosides, and straight-chain aliphatic compounds [179]. The extracts of this plant have been studied for their anti-inflammatory, antibacterial, and anticancer properties. Rabi et al. found that Aphanin, one of the isolated novel triterpenoid compounds, exhibited antiproliferative effects, caused G_0_-G_1_ cell cycle arrest, inhibited K-Ras G12D mutant activity, and induced apoptosis in PC HPAF-II cells [91].

#### 3.5.4. (*Ancistrocladaceae*) liana

*Ancistrocladaceae* is the name of flowering plants growing in the tropics. The active ingredient is michellamine B, an isoquinoline alkaloid that can be found in the mature leaves. A number of other alkaloids have also been isolated that display strong cytotoxic activities against PC cell lines in nutrient-deprived media, without toxicity in normal, nutrient-rich conditions [180]. Li et al., using two newly discovered naphthylisoquinoline dimers, along with the known dimer jozimine A_2_, showed that they have strong cytotoxic activities against human PC cells under nutrition-deprived conditions [92].

#### 3.5.5. Apigenin

Apigenin (4′,5,7-trihydroxyflavone), a natural product found in many fruits and vegetables, mainly in the flowers of chamomile plants, belongs to the flavone class. Johnson et al. investigated the inhibitory effects of the citrus fruit bioactive compounds, namely flavonoids, limonoids, phenolic acids, and ascorbic acid, on human PC cells. Apigenin showed strong anticancer activity through the induction of pancreatic cell death arrest of the cell cycle at the G2 /M phase, and activation of the mitochondrial pathway of apoptosis. Apigenin could also up-regulate the expression of a number of cytokine genes in BxPC-3 cells [93]. Moreover, apigenin has been shown to sensitize PC cells to chemotherapy by affecting a number of molecular pathways [94].

#### 3.5.6. *Asteraceae* and *Lamiaceae*

The *Lamiaceae* or *Labiatae* is a family of aromatic flowering plants that includes herbs such as basil, mentha, rosemary, sage, savory, marjoram, oregano, thyme, and lavender. These plants contain phenolic compounds, especially flavonoids [181]. The effects of different *Asteraceae* (Achillea millefolium and Calendula officinalis) and *Lamiaceae* (Melissa officinalis and Origanum majorana) plant extracts against human PC cells were tested. In an experimental study, *Asteraceae* extracts showed significant antitumor activity by inducing cytotoxicity, and inhibiting cell transformation [95].

#### 3.5.7. Bitter Melon Juice

Bitter melon juice contains a large number of phytochemicals, flavonoids, triterpenes, saponins, ascorbic acid, steroids, proteins, and polysaccharides. Kaur et al. showed that bitter melon juice decreased the viability of PC cell lines by inducing apoptotic death. At the molecular level, it caused caspase activation, altered Bcl-2 expression, and decreased 22-signaling and X-linked inhibitor of apoptosis protein levels. The oral administration of bitter melon juice in mice inhibited MiaPaCa-2 tumor xenograft growth by 60% without side-effects [96].

#### 3.5.8. BRM270

BRM270 is a natural compound used in Asian traditional medicine, made from seven herbal plant extracts (Saururus chinensis, Citrus unshiu Markovich, Aloe vera, Arnebia euchroma, Portulaca oleracea, Prunella vulgaris var. lilacina, and Scutellaria bacicalensis). BRM270 is considered to be the most important phytochemical extract that possesses anticancer properties, although its low bioavailability makes the administration of high dosages necessary in order to obtain positive results. Huynh et al. investigated the effect of BRM270 on the isolated surface market CD44 positive PC cells. They showed that BRM270 induced apoptosis in these cells, and inhibited metastasis traits via the sonic hedgehog signaling pathway. Moreover, in an in vivo experiment, tumor growth derived from CD44^+^ PDAC was suppressed [97]. It seems that the administration of this phytochemical extract selectively targeting CD44^+^ PDAC cells in tumors might be an effective approach against pancreatic tumorigenesis. BRM270 also has the advantage of being effective in PC cells resistant to paclitaxel and gefitinib [98].

#### 3.5.9. *Boesenbergia pandurata*

*Boesenbergia rotunda* is a medicinal herb growing mainly in China and Southeast Asia. Nearly a hundred compounds were isolated and elucidated, includung flavonoid and chalcone derivatives, esters, kawains, terpenes, and terpenoids. It is specifically used as a spice, or as a flavoring agent, or in traditional medicine in patients with liver cirrhosis. Its rhizomes have been used in various disorders, including oral eczema, ulcers, and dry mouth. Its extracts improve candidiasis, especially in patients with HIV infection, dysentery, and abdominal pain. A recently described observation, according to which human PC cell lines exhibit a significant tolerance to nutrition starvation agents that could inhibit the survival of cancer cells under low nutrient conditions, might be a very interesting novel strategy in the treatment of PC. In this regard, Nguyen et al. tested an extract of the rhizomes of *Boesenbergia pandurata* against human PC cells under nutrient-deprived conditions. They showed that the isolates isopanduratin A1 and nicolaioidesin C exhibited the strongest cytotoxicity against human PC cells under nutrition-deprived conditions [99].

#### 3.5.10. *Boswellia sacra* Gum Resins

Gum resins from *Boswellia* species have been used in Ayurvedic and Chinese medicine in a number of clinical applications, including for malignant disorders. Ni et al. showed that crude essential oil prepared from hydrodistillation of *Boswellia sacra* gum resins could reduce viability, and increase cancer cell death. Human PC cells showed reduced viability, and increased death after treatment with fractions III and IV. *Boswellia sacra* essential oil Fraction IV also exhibited anticancer activities against PC in the heterotopic xenograft mouse model [100]. Although the responsible chemical compounds were not identified, *Boswellia sacra* gum resins are promising agents for the treatment of PC.

#### 3.5.11. *Bruceine D*

*Brucea javanica* is a shrub belonging to the family *Simaroubaceae*, growing naturally in a number of countries, including Sri Lanka and India to China, Malesia, New Guinea, and Australia. The fruit *Brucea javanica* contains quassinoid compounds that have anticancer and antiparasitic properties. *Brucea javanica* oil emulsion has been employed as adjunctive therapy for the treatment of various cancers. The mechanisms of antitumor activity may include inhibition of DNA polymerase activity, arrest of the tumor cell division cycle, disruption of the cellular energy metabolism, and depression of the expression of vascular endothelial growth factor. Liu et al. showed that *bruceine D* has a potent cytotoxic effect on PC cell lines. *Bruceine D* induces cytotoxicity in Capan-2 cells via the induction of cellular apoptosis involving the mitochondrial pathway. The antiproliferative effects of *bruceine D* were comparable to those exhibited by camptothecin and gemcitabine. Finally, the expression of both caspase 9 and caspase 3 in BD-treated Capan-2 cells was accentuated [101]. *Brucea javanica* oil emulsion can effectively reverse the multidrug resistance of tumor cells, thus increasing the sensitivity of cancer cells to chemotherapy and radiotherapy. Yang et al. found that the combination of Brucea javanica with gemcitabine in a PC patient-derived orthotopic xenograft mouse model resulted in a reduced tumor growth rate, and increased apoptosis compared to the vehicle control and gemcitabine alone, as well as increased survival. Taken together, all of these results unequivocally indicate that this plant has a therapeutic potential for PC [102].

#### 3.5.12. Cannabinoids

Cannabinoids are chemicals found in cannabis. Phytocannabinoid tetrahydrocannabinol represents the primary psychoactive compound of cannabis. There are at least 113 different cannabinoids isolated from cannabis, exhibiting varied therapeutic effects, including improvement of the outcome of cancer patients. In vitro studies have shown that cannabidiol has antiproliferative and proapoptotic effects mediated through cannabinoid receptor-1 and -2, and G-protein-coupled receptor 55 pathways. In vitro studies with cannabidiol, tetrahydrocannabinol, and synthetic derivatives also demonstrated tumor growth-inhibiting effects. The combination of cannabidiol cannabidiol/synthetic cannabinoid receptor ligands and chemotherapy in xenografts showed positive results [103].

#### 3.5.13. *Citrus Unshiu* Peel

*Citrus Unshiu*, 1 of the more than 900 citrus species known today, is a seedless, easy to peel tangerine, coming from the Japanese town, Satsuma. The chemical constituents of the *citrus unshiu flower* include γ-terpinene (24.7%), 2-β-pinene (16.6%), 1-methyl-2-isopropylbenzene (11.5%), L-limonene (5.7%), β-ocimene (5.6%), and α-pinene (4.7%) [182]. Lee et al. prepared a fermented extract of Citrus unshiu peel from the byproduct after juice processing, and examined its anticancer effects on PC cells. They found that fermented *Citrus unshiu peel* mainly consists of aboriginal compounds (narirutin and hesperidin), as well as newly generated compounds (naringenin and hesperetin). Treatment with fermented *Citrus unshiu* peel inhibited the growth of human PC cells through the induction of caspase-3 cleavage both in vitro and in vivo. Moreover, *Citrus unshiu peel* also blocked the migration of the PC cells through the activation of intracellular signaling pathways [104]. Naringenin and hesperetin were the unique modules related to its anticancer effect. In in vivo xenograft models, fermented Citrus unshiu peel also showed anticancer effects, suggesting that this product might be an effective anticancer drug for PC with no side-effects.

#### 3.5.14. Cloves (*Syzygium aromaticum*)

Cloves are the aromatic flower buds of a tree in the family Myrtaceae, *Syzygium aromaticum*. They are native in Indonesia, used as a spice. Clove oil containing eugenol could be effective in pain of various origins, although no supporting data exist. Again, inconclusive results were obtained from studies investigating its efficacy in fever and diabetes. Therefore, the use of cloves for any medicinal purpose has not been approved by the FDA of the USA. Some side-effects have also been described in patients suffering from liver, and blood clotting and immune system disorders. However, in the study of Li et al., the aqueous extract of cloves inhibited cancer cell growth, and diminished the colony formation on several cancer cell lines, including human PC cells. An in vivo study revealed that aqueous extract of cloves inhibited the tumor growth in a HT-29 xenograft mice model by inducing cell autophagy [105].

#### 3.5.15. Cocoa Polyphenol

Epicatechin is the major polyphenol contained in cocoa. Siddique et al. showed that treatment with cocoa polyphenol and epicatechin decreased the NF-κB transcriptional activity of premalignant and malignant Kras-activated pancreatic ductal epithelial cells, concurrently decreasing their proliferation, the guanosine triphosphate-bound Ras protein, and the Akt phosphorylation [106]. Furthermore, the oral administration of 25 mg/kg of cocoa polyphenol inhibited the growth of *Kras*-PDE cell-originated tumors in a xenograft mouse model.

#### 3.5.16. Cordyceps Militaris

Cordyceps militaris is a species of fungus in the family Cordycipitaceae. It represents an entomopathogenic fungus, which is widely used in traditional Chinese medicine as a general booster for the nervous system, metabolism, and immunity. A variety of substances, including saccharides, nucleosides, mannitol, and sterols, have been isolated from this fungus. The biological activity of Condyceps militaris is attributed to the saccharide and nucleoside contents. In a recently published study, Li et al. demonstrated that Cordycepin inhibits PC growth in vitro and in vivo via targeting FGFR2, and blocking ERK signaling [107].

#### 3.5.17. Crocus Sativus

*Crocin*, the main component of Crocus sativus, is one of the few water soluble carotenoids found in nature [183]. In an experimental study, it was shown that *Crocin* induced apoptosis and cell cycle arrest of BxPC-3 human PC cell lines, and decreased cell viability. The mechanism of action seems to be related with the induction of apoptosis of the malignant cells [108].

#### 3.5.18. Cryptotanshinone

*Salvia miltiorrhiza* is a perennial plant of the genus *Salvia*. The chemical compositions of this herb are hydrophilic phenolic compounds and lipid-soluble diterpenoid compounds that are responsible for its pharmacological activities. The tanshinones in *Salvia miltiorrhiza* are diterpenoid quinones, which can be classified into two series, one is the phenanthro [1,2-b] furan-10,11-diones; and the other is the phenanthrol [3,2-b] furan-7, 11-diones. Alone or combined with other herbal medicines, *Salvia miltiorrhiza* has been used in China as a treatment for cardiovascular and cerebrovascular diseases, although a number of meta-analyses suggest that the effects of the plant are inconclusive. Adverse effects may include allergic reactions, dizziness, headache, gastrointestinal complaints, and bleeding in patients taking warfarin. *Danshen* may have anti-hypertensive and anti-platelet aggregation effects, and may enhance endogenous anti-oxidative enzyme activities. Cryptotanshinone, one of the active constituents of *Salvia miltiorrhiza*, has been shown to exhibit significant antitumor effects in several cancer cells [109]. Ge et al. demonstrated that cryptotanshinone inhibited proliferation, and induced cell apoptosis and cell cycle arrest in PC cells. In addition, cryptotanshinone decreased the activities of signal transducer, as well as several upstream regulatory signaling pathways [110].

#### 3.5.19. *Cucurbitacin E*

*Cucurbitacin E* belongs to a triterpenoid family, isolated from plants, and shows antiproliferative activity on various cancer cells. *Cucurbitacin E* inhibited the growth of PANC-1 cells and apoptosis, inhibited STAT3 phosphorylation, and up-regulated p53 expression [111].

#### 3.5.20. *Cucurmosin*

The structure of *Cucurmosin* has been proved to be one of the type 1 ribosome-inactivating proteins. Xie et al. established an NOD-SCID mice orthotopic transplantation model, and estimated the proliferation inhibition effect of *Cucurmosin* in SW-1990 cells in vivo. *Cucurmosin* inhibited the proliferation of PC cells, and induced apoptosis in a dose- and time-dependent manner. In the NOD-SCID mice models, the tumor proliferation inhibition rates were increased compared with controls. Cucurmosin inhibited the examined proteins in the PI3K/Akt/mTOR signaling pathway, and induced active fragments of Caspase 3 and 9 [112]. Cucurmosin can inhibit the growth and induce apoptosis of the human PC cell line SW-1990 both in vitro and in vivo.

#### 3.5.21. *Dandelion root* Extract

*Dandelion root* has been used in traditional Chinese and Native American medicine in various liver and gastric disorders. The therapeutic properties of *Dandelion* are attributed to its bioactive chemical components, including chicoric acid, taraxasterol, chlorogenic acid, and sesquiterpene lactones [184]. In an experimental study, *Dandelion root* extract induced selective apoptosis, as well as collapse of the mitochondrial membrane potential, leading to prodeath autophagy [113].

#### 3.5.22. *Degalactotigonin*

*Solanum nigrum* is a species in the genus *Solanum*, native to Eurasia, and introduced in America, Australasia, and South Africa. *Solanum nigrum* is a widely used plant in oriental medicine, considered to have anticancer, antioxidant, anti-inflammatory, hepatoprotective, diuretic, and antipyretic properties. However, ripe berries can cause symptoms such as fever, sweating, vomiting, abdominal pain, diarrhea, confusion, and drowsiness. Death after ingesting large amounts of the plant has been observed. *Degalactotigonin* solasodine a steroidal glycoside isolated from *Solanum nigrum* which showed cytotoxicity, and induced apoptosis in PC cell lines. It also inhibited EGF-induced proliferation and migration, and induced down-regulation of cyclin D1. Furthemore, it inhibited EGF-induced phosphorylation of EGFR, as well as activation of EGFR downstream signaling molecules [114].

#### 3.5.23. Diterpene 25 Signaling

Pyruvate dehydrogenase kinase 4 expression is up-regulated in various cancer tissues, being a suitable target for cancer therapy given its ability to shift glucose metabolism. Tambe et al. identified natural diterpene 25 signaling (KIS compounds) that inhibits pyruvate dehydrogenase kinase 4. They showed that KIS37 (cryptotanshinone) inhibited KRAS-activated human PC cell lines, and suppressed KRAS protein expression. Furthermore, KIS37 suppressed phosphorylation of Rb and cyclin D1 proteins, as well as the expression of cancer stem cell markers. KIS37 also suppressed PC cell growth in both subcutaneous xenograft and orthotopic pancreatic tumor models [115]. Therefore, KIS37 should be considered as a novel therapeutic strategy for targeting PDK4 in KRAS-activated PC.

#### 3.5.24. Elemene

Elemenes are a group of chemical compounds found in a variety of plants. Chemically, they are structural isomers of each other, and are classified as sesquiterpenes. They could be found in a variety of medical plants. Bearing in mind the antiproliferative effects of this compound, Long et al. investigated different doses of elemene in mice undergoing subcutaneous xenograft with BxPC-3 PC cells. In the in vitro experiment, a significant antiproliferative effect of BxPC-3 and Panc-1 cells was observed. In the in vivo BxPC-3 xenografts, elemene decreased the tumor size, up-regulated the expression of P53, and down-regulated the expression of Bcl-2 in the tumor [116]. The mechanisms of action against PC are related to down-regulation of the expression of Bcl-2, and up-regulation of the expression of P53.

#### 3.5.25. *Ellagic acid*

*Ellagic acid* is a natural polyphenol antioxidant found in a number of plants and fruits. This chemical substance is considered to be a dietary supplement with antineoplastic characteristics. However, *Ellagic acid* has been characterized by the USA FDA as a “fake cancer cure’”. It has been suggested that urolithin A, microflora metabolites of dietary ellagic acid derivatives, might have anticancer effects. Zhao et al. showed that treatment of PANC-1 xenografted mice with *Ellagic acid* resulted in a significant inhibition in tumor growth; suppression of cell proliferation and caspase-3 activation; induction of PARP cleavage; inhibition of the expression of Bcl-2, cyclin D1, CDK2, and CDK6; and induction of the expression of Bax in tumor tissues. Other effects included the inhibition of angiogenesis and metastasis in tumor tissues [117]. In a more recent study, Cheng et al. found that *Ellagic acid* significantly inhibited human PC PANC-1 cell growth, cell repairing activity, and cell migration and invasion. On the other hand, treatment of PANC-1 xenografted mice with *Ellagic acid* resulted in a significant inhibition in tumor growth, and prolongation of the mice survival rate. [118]. The use of *Ellagic acid* would be beneficial for the management of PC.

#### 3.5.26. *Emodin*

*Emodin* (1, 3, 8-trihydroxy-6-methylanthraquinone), a Chinese medicinal herb with antineoplastic action, represents the active constituent isolated from the root of *Rheum palmatum L.* In a study aiming to investigate its antineoplastic action, Liu et al. noticed that *emodin* induced significant growth inhibition and apoptosis in the PC cell line SW1990 compared to that of control, concurrently suppressing the migration and invasion of SW1990 cells. *Emodin* down-regulated the NF-κB DNA-binding activity in SW1990 cells, and up-regulated the expression of caspase-3. In addition, the oral administration of *emodin* decreased tumor weight and metastasis [119].

#### 3.5.27. *Eryngium billardieri*

*Eryngium* represents a genus of flowering plants belonging to the family *Apiaceae*, with a global distribution. Eryngium contains several chemical constituents, including sesquiterpenes and monoterpenes as the main components, as well as aldehydes, coumarins, sitosterols, and sugars. They are annual and perennial herbs with spiny leaves. Some species are native to rocky and coastal areas, but the majority of them are grassland plants. Many species of *Eryngium* have been used in food and medicine against diabetes mellitus, and as an antiinflammatory agent [120]. Roshanravan et al. have shown that *Eryngium billardieri* extracts had cytotoxic effects on PANC-1 cancer cell lines, and induced apoptosis [121]. Moreover, treatment of cancer cells with dichloromethane and n-hexane extracts of *Eryngium billardieri* induced the overexpression of Bax and underexpression of cyclin D1.

#### 3.5.28. Eucalyptus

Eucalyptus is a genus of over seven hundred species of flowering trees, shrubs, or mallees in the myrtle family commonly known as eucalypts. Although eucalyptus mainly grows in Australia, other species are now distributed globally. Being a natural insecticide, it can indirectly reduce the risk of malaria. Eucalyptus oils have been used in the pharmaceutical and cosmetics industries for multiple purposes. It has been used for the treatment of flu, fever, muscular aches, sores, pains of various origin, and inflammation. Eucalypts have been linked with cytotoxic and anticancer properties, although little scientific evidence exists [122]. Bhuyan et al. assessed the anticancer properties of aqueous and ethanolic extracts of four Eucalyptus species, and found these exctracts inhibited the growth of PC cells by more than 80% at 100μg/mL. Caspase 3/7-mediated apoptosis and morphological changes of cells were also witnessed in MIA PaCa-2 cells [123]. In a subsequent study [124], the same group of investigators found that *Angophora floribunda* extract exerted a greater cell growth inhibition followed by A. hispida in MIA PC-2 cells.

#### 3.5.29. Ferula Hezarlalehzarica

The genus Ferula comprises more than 170 species worldwide, of which 30 grow in Iran. These plants have been used traditionally for treating skin infections and stomach disorders. So far, there is a lack of phytochemical investigations of this plant. Alilou et al. isolated and evaluated 18 compounds, and found that the dichloromethane extract of the roots of Ferula hezarlalehzarica is a rich source of bioactive compounds for targeting PANC-1 cells [125].

#### 3.5.30. *Gallic acid*

In a relevant study, a grape seed *procyanidin extract* significantly inhibited cell proliferation, and increased apoptosis in PC cells through down-regulation of the antiapoptotic protein Bcl-2, depolarization of the mitochondrial membrane, and reduction of the formation of reactive oxygen species. *Gallic acid* had the highest antiproliferative and proapoptotic activity [126].

#### 3.5.31. Garlic

Garlic (*Allium sativum*) represents a species in the onion genus, *Allium*. An inverse association between garlic intake and gastric cancer was previously suggested. A number of side-effects, including gastrointestinal discomfort, sweating, dizziness, allergic reactions, bleeding, menstrual irregularities, and bad breath (halitosis), have been described. Wang et al. investigated a novel garlic active component (S-propargyl-L-cysteine), and showed that this substance reduced cell viability and colony formation, inhibited cell proliferation, and induced G2/M phase cell cycle arrest and apoptosis in human PC cells. It also inhibited tumor growth in Panc-1 xenografts by regulating the JNK protein levels [127]. Lan et al. [128] observed morphologic changes of PC cells under transmission electron microscopy after treatment with garlic oil for 24 h. In the higher garlic oil concentrations, an earlier change of the apoptotic tendency was detected.

#### 3.5.32. *Gedunin* (*Azadirachta indica*)

*Gedunin* is a tetranortriterpenoid isolated from the Indian neem tree *Azadirachta indica*. It has been used for the treatment of malaria and other infectious diseases in traditional Indian medicine. In addition, *gedunin* has demonstrated antiproliferative activity against various cancer cell lines. Subramani et al. assessed the anti-metastatic potential of *gedunin* on PC cells using matrigel invasion, cratch, and soft agar colony formation assays. They found that *gedunin* treatment was highly effective in inducing the apoptosis of PC cells. Furthermore, *gedunin* inhibited metastasis of PC cells by decreasing their invasive, migratory, and colony formation capabilities [129].

#### 3.5.33. Ginger Extract

Ginger is a plant native to warmer parts of Asia, but now is growing also in parts of South American and Africa. It is likely safe when taken appropriately, although it can cause mild side-effects, including heartburn, diarrhea, abdominal discomfort, and extra menstrual bleeding. Ginger spice comes from the roots of the plant. The extract of ginger and its major pungent components have an anti-proliferative effect on several tumor cell lines. Akimoto et al. demonstrated that the ginger ethanol-extracts suppressed cell cycle progression, and increased the death of human PC cell lines by inducing autosis, a form of cell death. Daily intraperitoneal administration of the extract prolonged survival in a peritoneal dissemination model, and suppressed tumor growth in an orthotopic model of PC [130].

#### 3.5.34. Ginkgolic Acid

Ginkgolic acid is a botanical drug extracted from the seed coat of *Ginkgo biloba* L., harboring anti-tumor effects. It has been used for centures in China and in Europe (Germany) some decades ago, in the treatment of Alzheimer’s disease. However, *Ginkgo biloba* leaf extracts may have undesirable effects in patients taking anticoagulants. Other side-effects include nausea, vomiting, diarrhea, headache, dizziness, and heart palpitations. It should be avoided during lactation. Ma et al. showed that *Ginkgo biloba* leaf extracts reduced the viability of cancer cells without toxic effects on normal cells. *Ginkgo biloba* leaf extracts also impaired colony formation, migration, and invasion ability, and increased apoptosis of cancer cells through the activation of AMP-activated protein kinase signaling and down-regulation of the expression of key enzymes involved in lipogenesis [131].

#### 3.5.35. Grape Proanthocyanidin

It has been found that in vitro treatment of human PC cells with proanthocyanidins from grape seeds resulted in a significant reduction in the cell viability, as well as a significant increase of G2/M phase arrest, induction of apoptosis, decrease in the levels of Bcl-2 and Bcl-xl, and increase in the levels of Bax and activated caspase-3. On the other hand, in vivo experiments using diet supplementation with proanthocyanidins from grape seeds on Miapaca-2 pancreatic tumor xenografts grown subcutaneously in athymic nude mice resulted in reduction of tumor growth, increased expression of Bax, reduction of anti-apoptotic proteins, and activation of caspase-3-positive cells [132].

#### 3.5.36. Graviola

The tropical tree *Annona Muricata*, commonly known as graviola, has been shown to have some anticancer properties. More than 200 chemical compounds have been identified and isolated from this plant; the most important being alkaloids, phenols, and acetogenins [185]. Torres et al. showed that graviola extract induced necrosis of PC cells by inhibiting cellular metabolism, and down-regulating the expression of NFκB [133]. In vitro functional assays confirmed the inhibition of growth of PC cells, suggesting that graviola extract is able to inhibit multiple signaling pathways regulating metabolism, survival, and metastatic potential of PC cells.

#### 3.5.37. Green Tea Extract

Green tea polyphenols have been shown to exhibit multiple antitumor activities in various cancers. Zhang et al. showed that green tea extract inhibited molecular chaperones heat-shock protein 90, its mitochondrial localized homologue Hsp75, and heat-shock protein 27, concomitantly. Furthermore, green tea extract inhibited Akt activation and the levels of mutant p53 protein, and induced apoptosis and growth suppression of the cells. The authors of this study discovered new molecular targets of the green tea extract, and provided further evidence on the activity of green tea in PC [134].

#### 3.5.38. *Helicteres hirsuta Lour*

*Helicteres hirsuta Lour* is the name of a herbal medicine used for the treatment of malaria and diabetes. Terpenoids, flavonoids, and lignans are the dominant constituents of Helicteres species [186]. Pham et al. showed that the leaf and stem extracts from *Helicteres hirsuta* and their aqueous and saponin-enriched butanol sub-fractions possessed a strong anticancer activity in vitro against MIA PaCa-2, BxPC-3, and CFPAC-1 cells [135].

#### 3.5.39. *Inula helenium*

*Inula helenium* is a widespread plant species belonging in the sunflower family *asteraceae*. It is native in Eurasia, Spain, and Xinjiang Province in western China. The constituents of *Inula helenium* are sesquiterpene lactones eudesmanolides, having anticancer, anti-inflammatory, antimicrobial, antiproliferative, and cytotoxic properties. Sesquiterpenoids contain thousands of compounds, and have been described as the active components of various medicinal plants used in traditional medicine. Zhang et al. showed that low concentrations of the extract of *inula helenium* caused cfpac-1 cell cycle arrest, whereas high concentrations induced mitochondria-dependent apoptosis. In addition, ethyl acetate extract of *inula helenium* inhibited the phosphorylation of the signal transducer and activator of the transcription (stat)3/akt pathway [136].

#### 3.5.40. *Lonicera japonica*

*Lonicera japonica* is a species of honeysuckle native to eastern Asia, and has the highest contents of chlorogenic acid [187]. The dried leaves and flowers are used in traditional Chinese medicine to treat fever, cough and thirst, and certain inflammatory disorders, including viral infections. Lin et al. investigated a homogenous polysaccharide extracted and purified from flowers of *Lonicera japonica* in PC cell lines. They showed that *Lonicera japonica* inhibited BxPC-3 and PANC-1 PC cell growth at the concentration of 1 mg/mL [137].

#### 3.5.41. *Lupeol*

*Lupeol* is a dietary triterpene present in many fruits and medicinal plants. This ingredient possesses a number of pharmacological properties, including in vitro and in vivo anti-cancer activities. Liu et al. [138] showed that *Lupeol* inhibited the proliferation of PC cells, and induced apoptosis and cell cycle arrest.

#### 3.5.42. *Mangifera indica*

*Mangifera indica*, commonly known as mango, is a species of flowering plant in the sumac and poison ivy family *Anacardiaceae*. Mango components can be grouped into macronutrients (carbohydrates, proteins, amino acids, lipids, fatty, and organic acids), micronutrients (vitamins and minerals), and phytochemicals (phenolic, polyphenol, pigments, and volatile constituents). Mango fruit also contains structural carbohydrates, such as pectins and cellulose. The most important organic acids include malic and citric acids [188]. It has been shown that different parts of the mango tree have different therapeutic properties. Nguyen et al. found that a methanol extract of the bark of *Mangifera indica* could inhibit the survival of human PC cells under nutrient-deprived conditions without an apparent toxicity. This methanol extract was found to consist of 19 compounds [139].

#### 3.5.43. Mexican Lime (*Citrus aurantifolia*)

*Lime* (*Citrus aurantifolia Swingle*) is a citrus fruit, having antineoplastic properties. Thirty-three chemical compounds were identified, with d-limonene forming the major constituent. Other prominent constituents include 3,7-dimethyl-2,6-octadien-1-ol geraniol *E*-citral *Z*-citral and β-ocimene [189]. Patil et al. extracted the bioactive compounds of Mexican lime (neohesperidin, hesperidin, and hesperitin) using different solvents. They showed that limonoids identified (limonexic acid, isolimonexic acid, and limonin) inhibited PC-28 growth. Furthemore, the induction of apoptosis was confirmed by the expression of Bax, Bcl-2, casapase-3, and p53, indicating that antioxidant activity depends on the flavonoids, whereas the inhibition of proliferation depends on the content of both flavonoids and limonoids [140].

#### 3.5.44. *Moringa Oleifera*

It has previously been described that the combination of *Moringa oleifera* with radiation excibits an additional inhibitory effect by overcoming the radioresistance of PC cells [61]. In this study, the authors investigated the effect of aqueous *Moringa oleifera* leaf extract on cultured human PC cells. They found that this extract inhibited the growth of all PC cell lines, and enhanced the cytotoxic effect of cisplatin on PC cell lines [141]. *Moringa oleifera* leaf extract inhibits the growth of PC cells and the cells’ NF-κB signaling pathway, and increases the efficacy of chemotherapy with cisplatin in human PC cells.

#### 3.5.45. *Matrine*

*Matrine*, an alkaloid extracted from the Chinese herb of the genus Sophora flavescens, has exhibited a variety of pharmacological effects, including anti-proliferative and pro-apoptotic properties. *Matrine* and the related compound oxymatrine act as a nematicide against a variety of nematodes [142]. Liu et al. showed in in vitro assays, that matrine inhibited cell viability by down-regulating the expression of PCNA, and induced cell apoptosis by reducing the ratio of Bcl-2/Bax, up-regulating Fas, and increasing activation of caspases-8, -3, and -9. In the in vivo model, *matrine* inhibited tumor growth, and regulated the tumoral gene expression [190].

#### 3.5.46. Naringenin and Hesperetin Combined Treatment

*Citrus unshiu* is a seedless citrus species grown in Japan, Spain, central China, Korea, the USA, South Africa, and South America. The chemical constituents of the *Citrus unshiu* include γ-terpinene 2-β-pinene 1-methyl-2-isopropylbenzene L-limonene β-ocimene and α-pinene [191]. Lee et al. showed that the combined treatment of naringenin and hesperetin inhibited the growth and the migration of human PC cells compared to separate treatment with naringenin or hesperetin, through the induction of caspase-3 cleavage [143]. Moreover, combined treatment inhibited the phosphorylation of focal adhesion kinase and p38 signaling compared with separate treatment. In in vivo xenograft models, the combination treatment again showed an anti-growth effect.

#### 3.5.47. *Nerium oleander*

Oleandrin, a cardiac glycoside, exerts an anti-proliferative activity in in vitro malignant cells. PBI-05204, an extract of *Nerium oleander* containing oleandrin, was tested in a human PC orthotopic model. It was found that all the control mice exhibited tumors, whereas 25% of the mice treated for 6 weeks with PBI-05204 (40 mg/kg) showed dissectible tumors. PBI-05204 also markedly enhanced the antitumor efficacy of gemcitabine. This novel botanical drug exerts its potent antitumor activity through down-regulation of PI3k/Akt and mTOR pathways [144].

#### 3.5.48. Nimbolide

Nimbolide is a triterpenoid extracted from the flowers of *Azadirachta indica*, which is a tree native to the Indian subcontinent. It represents a major component in Siddha and Unani medicine. Products made from this tree have been used in India for their anthelmintic, antifungal, antidiabetic, antibacterial, antiviral, contraceptive, and sedative properties. Neem leaves have also been used for various skin disorders, such as eczema and psoriasis. In adults, short-term use is probably safe, although long-term use may be toxic for the liver and kidneys. Subramani et al. assessed the anticancer properties of nimbolide against PC. They showed that nimbolide induces the generation of reactive oxygen species, thereby regulating both apoptosis and autophagy in PC cells. Nimbolide-mediated reactive oxygen species generation inhibited proliferation and metastasis via mitochondrial-mediated apoptotic cell death. In in vivo experiments, nimbolide was effective in inhibiting PC growth and metastasis [145].

#### 3.5.49. Obacunone

Obacunone is one of the oxygenated triterpenoids found in rutaceae family. It has many biological actions, including antiproliferative activities against cancer cells. Chidambara et al. investigated the antiproliferative action of obacunone on cultured PC cells. The plant induced the inhibition of Panc-28 cell proliferation in a dose- and time-dependent manner, with a concurrent induction of apoptosis. The plant was able to up-regulate the expression of p53 and pro-apoptotic protein Bax, and down-regulate the Bcl2, NFκB, and Cox-2 [146].

#### 3.5.50. *Ocimum sanctum*

*Ocimum sanctum* (“Holy Basil”) has been used in traditional Indian medicine in a variety of clinical situations. The chemical composition of Holy Basil, also known as tulsi, is highly complex, containing many nutrients and other biologically active compounds, the proportions of which may vary considerably [192]. Shimizu et al. showed that extracts of Ocimum sanctum leaves can inhibit the proliferation, migration, invasion, and induce apoptosis of PC cells in vitro. Intraperitoneal injections of the aqueous extract of *Ocimum sanctum* inhibited the growth of orthotopically transplanted PC cells. Mice treated with *Ocimum sanctum* extracts exhibited up-regulation of E-cadherin, and induction of apoptosis, whereas genes that promote survival and chemo/radiation resistance were down-regulated [147].

#### 3.5.51. *Oleuropein*

*Olea europaea* L. leaves are an agricultural waste product with a high concentration of phenolic compounds, especially *oleuropein*. *Oleuropein* has a significant anti-proliferative activity. Goldsmith et al. showed that the extracts of the Corregiola and Frantoio varieties of *Olea europaea L*. leaves significantly decreased the viability of PC cell lines relative to controls [148].

#### 3.5.52. Olive Biophenols (Oleuropein, Hydroxytyrosol, and Tyrosol)

Olives contain more than 100 different biophenols, the most important being hydroxytyrosol and tyrosol, as well as their secoiridoid derivatives, verbascoside, lignans, and flavonoids. The main pharmacological studies reported so far have dealt with the antioxidant, anti-inflammatory, cardiovascular, immunomodulatory, gastrointestinal, respiratory, antimicrobial, anticancer, and chemopreventive properties of these biophenols. As far as their safety is concerned, these products are generally considered safe, although further studies are needed. The olive biophenols, oleuropein, hydroxytyrosol, and tyrosol, showed cytotoxicity towards cancer cells without affecting normal cells. In a recent study, the authors treated PC cells and non-tumorigenic pancreatic cells with oleuropein, hydroxytyrosol, and tyrosol. They found that oleuropein displayed selective toxicity towards MIA PC cells and hydroxytyrosol towards MIA PC and HPDE cells. Furthemore, oleuropein and hydroxytyrosol induced apoptosis in MIA PC cells [149].

#### 3.5.53. *Oridonin*

*Oridonin* is an organic heteropentacyclic compound isolated from the leaves of the herb *Rabdosia rubescens*. It is an organic heteropentacyclic compound, an enone, a cyclic hemiketal, a secondary alcohol, and an ent-kaurane diterpenoid (Figure 6). This traditional Chinese medicine substance has been shown long ago to exhibit antitumor effects.

Bu et al. found that *oridonin* induced apoptotic cell death in PC cells in a dose-dependent manner; the p53 being responsible for this antineoplastic action. *Oridonin* also increased the expression of p-p53 and p21 in the PC cells [150]. To further verify the anti-cancer effects of oridonin, Gui et al. showed that 105 miRNAs were differentially expressed in oridonin-treated BxPC-3 human PC cells, indicating that oridonin inhibits BxPC-3 cells through regulating the expression of miRNAs [151].

#### 3.5.54. *Paeonia lactiflora*

The chemical compound *aeoniflorin*, a monoterpene glycoside, represents one of the major constituents of an herbal medicine derived from *Paeonia lactiflora*. *Aeoniflorin* has been shown to exhibit antitumor effects in various cancer types. Li et al. showed that *Aeoniflorin* suppressed the growth of PC cell lines, and sensitized them to X-ray irradiation. *Aeoniflorin* treatment resulted in a reduction of cell proliferation, and an increase in the expression of the apoptotic protein Bax [152]. These results suggest that *Aeoniflorin* inhibits PC growth by up-regulating the tumor-suppressor gene HTRA3.

#### 3.5.55. Palm Oil Phenolics and PALM JUICE

Palm oil and its components are increasingly used in foods such as cooking oils, margarines, shortenings, and confectionery products. It contains 50% saturated fatty acids, 40% monounsaturated fatty acids, and 10% polyunsaturated fatty acids [193]. Oil palm phenolics have been shown to have anti-carcinogenic activities. By using two PC cell lines, Ji et al. showed that oil palm phenolics suppressed PC cell proliferation. Oil palm phenolic-induced apoptosis was associated with a decrease in Bcl-XL expressions, and increased cleaved caspase-3, caspase-9, and PARP expression, thus confirming the anti-tumor effects of these substances [153].

#### 3.5.56. *Paramignya trimera* Root and *Phyllanthus amarus*

During recent years, a medicinal plant rich in saponins (more than 500 mg EE/g dried sample) named Xao tam phan (*Paramignya trimera* Guillaum) has been used in cancer prevention and treatment [154]. *Phyllanthus amarus,* a small herb belonging to the family *Euphorbiaceae*, is used widely, especially in Indian Ayurvedic medicine, for a number of pathological conditions [155]. Nguyen et al. assessed the cytotoxic activity of extracts and fractions from the *Paramignya trimera root* and *Phyllanthus amarus* against two PC cell lines. The root of *Paramignya trimera root* and the whole plant of *Phyllanthus amarus* were used. The findings revealed an impressive cytotoxic capacity of the *Paramignya trimera root* extract against both PC cell lines in a range of concentrations, which was higher than those of gemcitabine. In contrast, the cytotoxic capacity of the *Phyllanthus amarus* extract was significantly lower than that of gemcitabine. The IC_50_ values of the *Paramignya trimera root* extract were lower than that of the *Phyllanthus amarus* extract [156]. *Paramignya trimera* root extract could be a source for the development of new drugs against PC.

#### 3.5.57. *Plumbagin*

*Plumbagin*, isolated from *Plumbago zeylanica*, possesses an anticancer activity. The structure of its active principle is similar to that of vitamin K [194]. *Plumbagin* inhibited the growth of Panc-1 and Bxpc-3 cells by inducing apoptosis through the mitochondria-related pathway and caspase cascades [157].

#### 3.5.58. *Pomegranate extract*

*Pomegranate extract* is a standardized whole-fruit extract of pomegranate, which is a fruit rich in polyphenols, organic acids, sugars, polysaccharides, and minerals [195]. This extract exhibits strong antioxidant activity, having anticancer properties also. Nair et al. used PANC-1 and AsPC-1 human PC cells to test the effects of *Pomegranate extract*. It was shown that *Pomegranate extract* induced cell cycle arrest, and inhibited cell proliferation in PANC-1 cells. It is of interest that *Pomegranate extract* was more effective in inhibiting the proliferation of PANC-1 cells than the clinically used dose of paclitaxel [158]. It is possible that unidentified phytochemicals are responsible for the inhibitory effect of *Pomegranate extract*.

#### 3.5.59. *Pulsatilla koreana*

*Pulsatilla koreana* has been used as a traditional medicine for the treatment of a number of pathological situations in Korea. Phytochemical studies demonstrated the presence of protoanemonin, deoxypodophyllotoxin, oleanane, and 33 upine-type triterpenoid saponins on *Pulsatilla koreana* roots [196]. Son et al. showed that SB365 (saponin D isolated from the root of *Pulsatilla koreana*) suppressed the growth and proliferation of human PC cell lines by inducing apoptosis through an increase in the levels of cleaved caspase-3, and a decrease in the Bcl-2 expression. SB365 exerted also a significant anti-angiogenic effect. Finally, SB365 inhibited tumor growth through the induction of apoptosis, and inhibition of angiogenesis in an in vivo mouse xenograft [159].

#### 3.5.60. *Quercetin*

*Quercetin* represents one of most effective biopolyphenols chemicals present in several plants that have antioxidative and anti-inflammatory actions. Quercetin has been used as an adjunctive drug to PC treatment, acting through inhibition of autophagy and apoptosis oxidative stress, and enhancing the sensitivity to chemotherapy agents [160]. Borska et al. demonstrated that *quercetin* exerted cytotoxic action on two neoplastic cell lines. In the EPP85-181RDB cell line, *quercetin* sensitized resistant cells to daunorubicin, suggesting that it could break the resistance of neoplastic cells to chemotherapy [161]. This synergistic effect might allow the reduction in the total dose of the antineoplastic drug, thus reducing the rate of possible treatment side-effects. It can be used, therefore, as a supplementary drug to patients with PC.

#### 3.5.61. *Radix Scutellariae*

*Radix Scutellariae* represents the dried root of *Scutellariae baicalensis Georgi*. This plant has been extensively used in several Asian countries as an effective antinflammatory agent. Its root, known as *Radix Scutellariae*, is the source of the Chinese medicine *Huang Qin* for various clinical disorders. The main compounds responsible for the biological activity of skullcaps are flavonoids. Six flavones have proven to be the major bioactive flavones existing in the forms of aglycones and glycosides [162]. In a recent study, Liu et al. investigated the mechanisms of the total flavonoid aglycone extracted from *Radix Scutellariae* in inducing autophagy and apoptosis in PC cells in vitro and in vivo. In the in vitro experiments, they showed that total flavonoid aglycone extracted exhibited an anti-tumor activity, and induced apoptosis and autophagy in PC cell lines through the PI3K/Akt/mTOR signaling pathway. In the in vivo studies, they showed that 150 mg/kg of total flavonoid aglycone extracted inhibited the BxPC3 tumor growth in immune deficient mice, and induced both apoptosis and autophagy [163].

#### 3.5.62. *Rhazya stricta*

*Rhazya stricta* Decne is an important medicinal species used in South Asia and Middle East. Some of its alkaloids have been reported to have anticancerous properties. The study of Shaer et al. showed that the crude alkaloids extract of Rhazya stricta significantly induced apoptotic death in PC cells. They observed a significant decrease in cell viability in a dose-dependent manner [197].

#### 3.5.63. *Salvia chinensis*

*Salvia chinensis* is an annual plant growing in China belonging to the *Labiatae plant* family. This herbal medicine has been used in the treatment of hepatitis, as well as in breast, liver, and gastric cancer. *Salvia chinensis* consists of more than fifty chemical constituents, in four classes, namely terpenoids, phenolic acids, flavonoids, and dibenzylcyclooctadiene lignans [164]. Zhao et al. demonstrated that Salvia chinensis induced potent cytotoxicity in the MiapaCa-2 human PC cells. Under fluorescence microscopy, morphological features of apoptosis in the PC cell lines following treatment with the extract were also detected [165].

#### 3.5.64. *Sedum sarmentosum Bunge*

*Sedum sarmentosum Bunge* extract, a traditional Chinese herbal medicine, contains multiple active chemical components, including tricin-7-O-b-dglucoside, isorhamnetin, quercetin, and kaempferide [198]. This herbal has been used in a variety of clinical disorders, including liver diseases and other inflammatory situations. It has recently been used to treat tubulointerstitial damage in kidneys following injury [199]. However, the role of this extract has never been tested in PC. In this regard, Bai et al. showed that the *Sedum sarmentosum Bunge* extract inhibited cell growth, accompanied by down-regulated expression of proliferating cell nuclear antigen, and increased cellular apoptosis in a mitochondrial-dependent manner. Moreover, *Sedum sarmentosum Bunge* extract induced p53 expression, and inhibited epithelial-mesenchymal transition through down-regulation of the proliferation-related hedgehog signaling pathway. In animal xenograft models of PC, *Sedum sarmentosum Bunge* extract suppressed the growth of pancreatic tumors [166].

#### 3.5.65. *Sugiol*

*Sugiol* is an abietane diterpenoid natural product acting as an antiviral and antineoplastic agent, and as an antioxidant and radical scavenger. Hao et al. evaluated the anticancer activity of sugiol, and observed that sugiol reduced the cell viability of human PC cells through reactive oxygen species-mediated alterations in mitochondrial membrane potential, ultimately leading to apoptosis. Sugiol also caused cell cycle arrest in the G2/M phase, and up-regulated the expression of Bax, with down-regulation of Bcl-2 expression, indicating that it could be a potent molecule against PC [167].

#### 3.5.66. TEOA (2a,3a,24-*thrihydroxyurs*-12-en-28-*oicacid*)

TEOA (2a,3a,24-*thrihydroxyurs*-12-en-28-*oicacid*) a traditional Chinese herbal medicine which represents a pentacyclic triterpenoid isolated from the roots of *A. eriantha* Benth, exhibiting anti-inflammatory and anti-cancer effects both in vivo and in vitro. Recently, Yang et al. demonstrated that this herbal inhibited the proliferation and migration of PC cells concurrently, causing the induction of mitochondrial dysfunction in PANC1 and SW1990 cells. Finally, TEOA induced autophagic cell death in PC cells by inactivating the ROS-dependent mTOR/p70S6k signaling pathway [168].

#### 3.5.67. *Toosendanin*

*Toosendanin* is a triterpenoid extracted from Melia toosendan Sieb et Zucc, used as a digestive tract-parasiticide in ancient China. *Toosendanin* has a marked antibotulismic effect both in vivo and in vitro. Finally, *Toosendanin* can induce apoptosis in several cell lines, and suppress the proliferation of various human cancer cells [169]. Pei et al. found that *toosendanin* suppressed the viability and growth, as well as the migration and invasion of PC cells. Furthermore, toosendanin repressed xenograft tumor growth in mouse PC models. The substance has no significantly toxic side-effects. *Toosendanin* inhibits PC cell growth by blocking the Akt/mTOR signaling pathway [170].

#### 3.5.68. *Tripterygium wilfordii*

*Tripterygium wilfordii* Hook F is a vine plant, used widely in China as a herbal medicine. The main bioactive ingredients of *Tripterygium wilfordii* may be alkaloids, which, may account for its pharmacological properties. This herbal has anti-inflammatory, anticancer, and antibacterial activities, as well as beneficial effects on immune disorders. Zhao et al. recently performed network pharmacology on *Tripterygium wilfordii Hook F* using Traditional Chinese Medicine Systems Pharmacology and Gene Cards databases [171]. They screened out 22 ingredients and 25 target genes associated with PC. They found that triptolide-plasminogen activator urokinase could represent a novel target for patients with PC.

#### 3.5.69. *Valtrate*

*Valtrate*, also known as valtric acid, is a novel epoxy iridoid ester isolated from the Chinese aromatic medicinal herb Valeriana jatamansi Jones, belonging in the class of organic compounds known as iridoids and derivatives [200]. It has been utilized for medicinal purposes in China and India for many years. Chen et al. showed that valtrate inhibited the growth of PC cells by inducing apoptosis and cell cycle arrest. Moreover, valtrate inhibited the tumor growth of the PC cell PANC-1 in xenograft mice by 61%. The underlying mechanisms include an increase of the expression of Bax, suppression of Bcl-2, c-Myc and Cyclin B1, inhibition of the transcriptional activity of Stat3, and decrease in the expression of Stat3 [172].

#### 3.5.70. *Xanthohumol*

*Xanthohumol* is a natural product found in the female inflorescences of *Humulus lupulus*, also known as hops. It is a member of the class of chalcones that is trans-chalcone substituted by hydroxy groups at positions 4, 2′, and 4′; a methoxy group at position 6′; and a prenyl group at position 3′ [201]. Jiang et al. showed that *xanthohumol* inhibited the growth of PC cells and their xenograft tumors by inducing cell cycle arrest and apoptosis via inhibition of phosphorylation of the signal transducer, activation of the transcription 3, and expression of its downstream-targeted genes cyclinD1 and Bcl-xL. Xanthohumol might be a promising therapeutic agent against PC. The STAT3 signaling pathway is its key molecular target [173].

#### 3.5.71. *Xao tam phan* (*Paramignya trimera*)

*Xao tam phan* (*Paramignya trimera*) is a traditional medicinal plant used in the treatment of a number of cancers in Vietnam. This plant contains saponins, phenolics, flavonoids, and proanthocyanidins equiv., 81.49 mg rutin equiv., and 58.08 mg catechin equiv. (per g dried extract, respectively) [202]. Chemical analysis of *Paramignya trimera* leaves showed that total phenolic, total flavonoid, proanthocyanidin, and saponin contents were gallic acid, protocatechuic acid, ellagic acid, rutin, and quercetin. Powdered extract of the *Paramignya trimera* leaf exhibited anti-proliferative capacity against PC cell lines, being higher than those of ostruthin and gemcitabine. *Paramignya trimera* leaf extract represents a rich source of phytochemicals that possess antioxidant and anti-proliferative activities against PC [174].

#### 3.5.72. *Xylaria psidii*

*Xylaria* is a genus of ascomycetous fungi growing on dead wood. It represents an important source of biologically active metabolites. Two compounds, isolated from *Xylaria psidii*, namely xylarione A and 5-methylmellein, exhibited cytotoxicity against PC cells, with features characteristic of apoptosis. The cell cycle distribution confirmed a cell cycle arrest at the sub-G1 phase. Flow cytometry analysis displayed a substantial loss of mitochondrial membrane potential by both the compounds [175]. The isolated compounds from *Xylaria psidii*, namely xylarione A and 5-methylmellein, may serve as potential therapeutic agents for PC.

#### 3.5.73. *Wikstroemia indica*

*Wikstroemia indica* is a small shrub with small flowers and toxic fruits. It can be found in forests and on rocky, shrubby slopes in central and southeastern China, Vietnam, India, and the Philippines. It is an herb that has been used for a long time in traditional Chinese medicine. Four compounds, namely daphnoretin, chrysophanol, myricitrime, and rutin, were purified from *Wikstroemia* indica [203]. It has some side-effects, including dizziness, blurred vision, nausea, vomiting, abdominal distension and pain, and diarrhea. This plant has been used in clinical practice as an antipyretic, detoxicant, and expectorant. Chang et al. evaluated the in vitro cytotoxicity against PC cell lines of 26 compounds isolated from the roots of *Wikstroemia indica*. Two compounds, namely 8 and 12, displayed preferential cytotoxicity in the nutrient-deprived medium, without causing toxicity in normal nutrient-rich conditions [176].

#### 3.5.74. Ziziphus Nummularia

Ziziphus nummularia is a thorny shrub, rich in bioactive molecules, and bountiful in bioactive molecules, including tannins, flavonoids, steroids, glycosides, and alkaloids. The leaves of this plant have been used in the treatment of a number of pathological situations, including cancer. In an experimental study, Mesmar et al. found [177] that treatment of human PC cells with ethanolic extract (100–300 μg/mL) of Ziziphus nummularia inhibited cell proliferation and angiogenesis, and down-regulated the ERK1/2 and NF-κB signaling pathways.

## 4. Discussion

PC represents one of the most lethal human malignancies. The treatment of the disease is multidisciplinary, and depends on the stage of cancer at diagnosis. The available therapeutic modalities include the combination of surgery, chemotherapy, chemoradiotherapy, and supportive care. During the last years, the type of multidisciplinary treatment of this tumor is rapidly changing, especially in cases of locally advanced disease. It seems that the number of therapeutic options in metastatic disease will be transformed in the near future, combining personalized medicine, innovative targets, immunotherapy, therapeutic vaccines, and adoptive T-cell transfer [204]. However, despite the promising results obtained after the introduction of the FOLFIRINOX regimen, and, subsequently, the nab-paclitaxel in combination with gemcitabine, the prognosis of this malignancy remains disappointing. It is worth mentioning that, so far, neither personalized medicine nor immunotherapy achieved quite favorable results. On the other hand, the value of immune checkpoint inhibitors, one of the promising therapeutic modalities, is, at the moment, questionable. A number of other treatments, including combinations of chemotherapy with immunotherapy and vaccines or T-cells modified with a chimeric antigen receptor, could be of value in the future [205]. Cancer immunotherapy might improve the effectiveness of other therapeutic options, which might together improve the prognosis of PC [206]. Furthermore, although therapies targeting the stroma are promising, they do not represent the standard care at the moment [207]. Finally, novel therapies targeting BRCA1/2 mutations, mismatch repair deficiencies, and NTRK1-3 fusions have achieved good results in clinical trials [208]. Another major problem concerning the chemotherapy applied is the gemcitabine resistance of PC cells, leading to poor clinical outcomes. Even today, the underlying mechanisms for the development of gemcitabine resistance remain unclear. However, chemoresistance might be a result of the interaction between PC cells, cancer stem cells, and the tumor microenvironment [209]. Moreover, transcription factors, including enzymes and signaling pathways, participated in the nucleoside metabolism, and are probably involved in the development of chemoresistance to gemcitabin.

The issue of the contribution of herbal preparations and plants in the treatment of various human pathological conditions has been “adopted” by a significant percentage of the population of many countries, including those of the Western world [210]. So far, the publications on PubMed using the key words “plants”, “herbals”, AND “treatment” amount to 267,318 articles, of which 10,996 are clinical trials, 1141 are meta-analyses, 8346 are randomized clinical trials, 24,740 are reviews, and 1747 are systematic reviews. Already, in the first 2 weeks of 2022, the number of relevant articles in the PubMed database exceeded 1200, which proves the lively interest of researchers and clinicians regarding the role of plants and herbs in various diseases. As mentioned before, various herbal and plants have been used for thousands of years to treat various diseases in many countries, including China, India, Japan, and Korea. In Western countries, this kind of treatment represents the so-called complementary and alternative medicine. Concerning PC, it seems possible that many natural products derived from herbals and plants could play an important role in the prevention and treatment of this malignancy as an adjunctive approach. It has been suggested that more than 60% of the current anticancer chemotherapeutic drugs used in clinical practice were initially developed from plants and herbals. Compared to standard chemo-radiotherapy, herbal treatment has many advantages, including the enhancement of the immune system and suppression of tumor progression, while also reducing the side-effects of chemo-radiotherapy. Also, individual chemical substances derived from plants and herbals may have anti-oxidative, anti-inflammatory, immunoregulatory, and antiproliferative properties. Whereas Western medicine mainly focuses on targeting specific malignant molecular mechanisms, complementary medicine employs a holistic approach [211]. A network meta-analysis of Chinese herbal injections combined with chemotherapy for the treatment of PC published in 2017 concluded that Chinese herbal administration could be beneficial for PC patients in improving performance status, and reducing the adverse drug reactions [212].

In the present review, we have identified 86 studies looking at the effects of 74 different herbals and plant derivatives that have been investigated in experimental models of PC cell lines and PC xenografts. It is quite impressive that almost all studies showed beneficial effects. We identified a small number of clinical studies investigating the role of adding plants or plant extracts in the regular chemotherapy in patients with advanced PC. So far, three plants were tested, all of which showed beneficial results concerning the survival rate and the disease-free survival rate. Two more case reports showed remarkable improvement of the patients. Regarding clinical studies exploring the effect of plant derivatives on PC stem cells, we have identified five relevant studies, all of which revealed that the plants inhibited the proliferation rate of the cancer stem cells, and increased the efficacy of chemotherapeutics used in PC patients. Concerning studies combining nanotechnology with herbals and plants aiming to target PC cells, we identified seven such experimental studies, all of which showed promising results. Among the herbals investigated, Scutellaria barbata gold nanoparticles and Parvifloron D-loaded smart nanoparticles showed results that should be further investigated in clinical trials. The side-effects were very few, thus making this technology a very attractive one for the treatment of PC. Finally, we identified a satisfactory number of articles referring to 22 different plants and herbals, looking at the cytotoxic effects of the co-administration of chemotherapeutic agents with plants. Again, a positive therapeutic result was obtained in all studies.

The great majority of these plants and herbals have been previously used for centuries in the treatment of cancer. As previously mentioned, the effectiveness of the herbal and plant preparations was quite satisfactory in all studies, being significantly better compared to placebo. Therefore, and because of their multiple biological properties, herbs and plants should be tested in patients with PC. However, as herbal preparations are mixtures containing a huge range of biological compounds, it is difficult to know which component offers the most beneficial pharmacological effect or clinical benefit. The determination of herb components, dosage, and course during or after herb treatment becomes a challenge for clinical employment.

Individual herbal and plants usually pose more than one cellular and molecular mechanism explaining their antineoplastic action. As indicated in Table 2, most of the herbals and plants exhibited their effect via the induction of cellular apoptosis involving the mitochondrial pathway, through p53- and caspase-dependent induction of p38 MAPK, (Figure 7) inhibition of BxPC-3 cells through the regulation of the expression of miRNAs, increase of caspase-3 and decrease of Bcl-2 expression, decreased expression of major factors of angiogenesis, and induction of apoptosis and autophagy through the PI3K/Akt/mTOR signaling pathway. Also, the inhibition of K*ras*-activated PC cell lines, suppression of K*ras* protein, suppression of phosphorylation of Rb and cyclin D1, suppression of the NF-κB signaling pathway (Figure 8), and PC cell growth in xenografts were observed.

There are some concerns regarding the use of herbal mixture extracts instead of using single natural products. It is possible that herbal mixture extracts work synergistically, although the existing scientific data are insufficient. The most important “side-effect” of the use of herbal preparations is the abandonment of the drugs used in the treatment of neoplastic diseases by the patients, a fact leading to a deterioration of the underlying disorder. Toxic products can be found in some plant preparations. The long-term safety of herbal treatment, including possible mutagenicity, has not been adequately explored.

## 5. Strengths and Limitations

The study included all clinical and experimental studies published in the international literature up to the middle of 2021. Depending on their nature, the studies were divided into those related to the effect of plants and herbals in patients with PC, and those related to their effect in PC cell lines. The studies related to the effect of plants and herbs using nanotechnology, as well as the studies related to the effect of herbs on PC stem cells, were also mentioned separately. The description was made in alphabetical order so as to facilitate the reader in finding and studying a particular plant or herb. A summary of information regarding the anticancer chemicals contained in every plant or herbal, as well as the underlying mechanism(s) of their action, were also mentioned. Certainly, there are limitations of our study, which are related to: (i) the inability to describe all clinical and experimental data due to their vast amount; (ii) only including studies published in the English language; (iii) being unable to rule out the possibility that plant dosages and herbals used in the experimental and/or clinical studies may be different from the optimum ones; and (iv) the side-effects of these herbal products not being mentioned in detail, but only including the most important of them. Because these natural products have an extremely complex chemical composition on the one hand, and, on the other, they act at different levels, further studies on their bioavailability and exact dosage will be needed. We believe that, in the future, the main use of these herbal products should be tested at a clinical level, administered in combination with conventional chemotherapy, and especially in cases of resistance to chemotherapy.

## 6. Conclusions

The popularity and acceptability of herbal medicine in developed countries, as well as its availability, safety, and low cost are steadily increasing. According to WHO estimations, 60% of the world’s population, and 80% of the population in developing countries are using herbal medicine [210]. However, some concerns related to their use with respect to their pharmacognosy and standardization compared with conventional drugs have been raised. Certainly, an improvement in the methods used for the categorization, storage, and quality control of these compounds is necessary. It should be emphasized that the somewhat positive effect of plants and herbs is met with skepticism and reservation by some members of the scientific community. We believe that due to the limited number of clinical trials assessing the role of herbals and plants, and especially their interaction with conventional chemotherapy and polypharmacy in PC patients, caution is required and more clinical studies are needed before affirming that herbals and plants can be effective strategies in PC, and that alternative medicine should enter medical education. Indeed, most studies have been performed in the preclinical setting, and, at the present time, there is insufficient data to draw conclusions on the clinical management of PC patients.

In summary, we can assume that: (i) the number of studies referring to plants and herbals was very large; (ii) all experimental studies confirmed that the plants tested exhibited anticancer potential; (iii) the few clinical studies in which plants were used as a complementary treatment to chemotherapy increased the clinical effect of the chemotherapeutic agent; (iv) all the authors of the relevant papers agree that large clinical trials using different plants as complementary agents to chemo-radiotherapy should urgently be performed. Cancer chemotherapy is very expensive. Although the cost of plant therapy seems to be very low, we suggest that future studies should also include details concerning the exact cost of this therapy. Pharmaceutical companies must contribute to the current knowledge by supporting relevant studies. International scientific societies and governmental organizations should seriously take into account the locally available opportunities of drug development by supporting clinical studies. With a discerned safety of herbs and plants, this kind of treatment, either alone or, preferably, in combination with conventional therapies, would largely benefit patients with pancreatic neoplasms. We suppose that reporting evidence-based scientific observations and case reports might encourage the pharmaceutical companies and government agencies to undertake large scale clinical trials to assess the long-term safety and efficacy of alternative anticancer therapies in humans.

## Figures and Tables

**Figure 1 nutrients-14-00619-f001:**
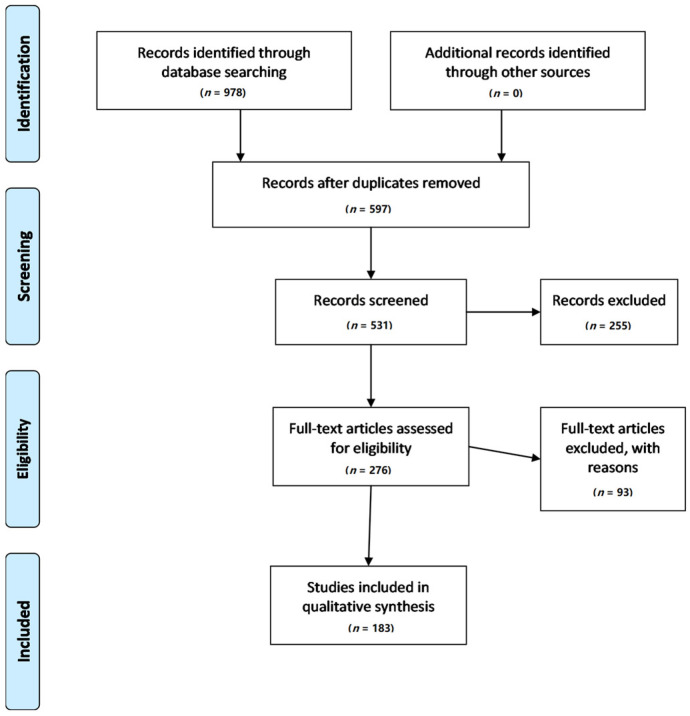
PRISMA flow diagram used in this systematic review.

**Figure 2 nutrients-14-00619-f002:**
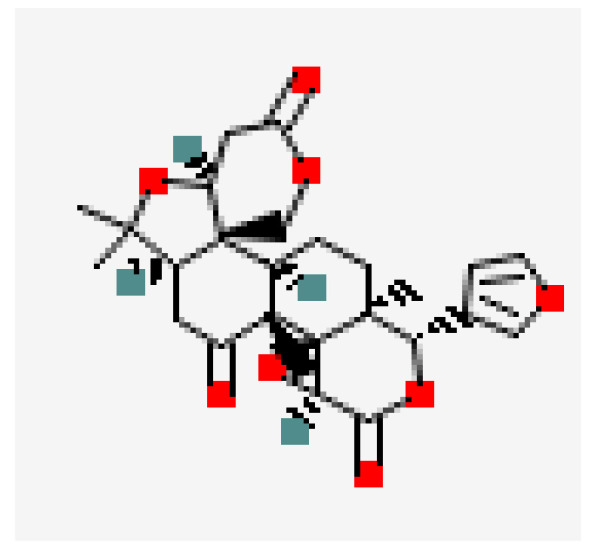
Chemical structure of Limonin (Available online: https://pubchem.ncbi.nlm.nih.gov, accessed on 10 December 2021).

**Figure 3 nutrients-14-00619-f003:**
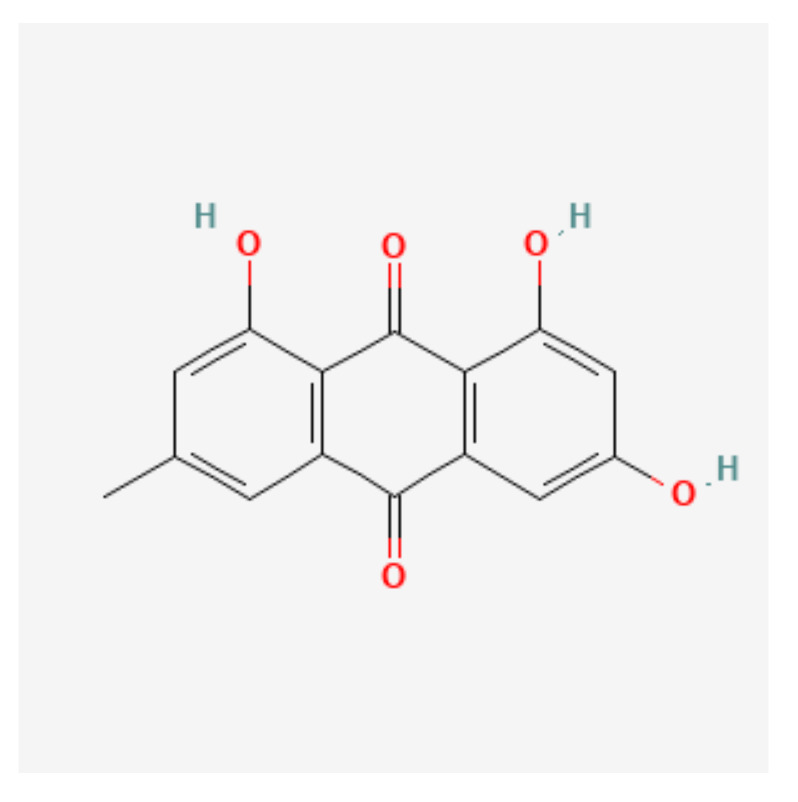
Chemical structure of Emodin Available on line https://pubchem.ncbi.nlm.nih.gov, (accessed on 10 December 2021).

**Figure 4 nutrients-14-00619-f004:**
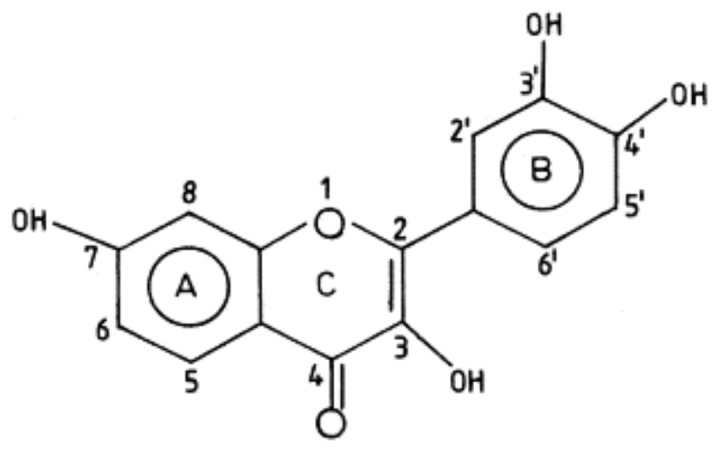
Chemical structure of Fisetin.

**Figure 5 nutrients-14-00619-f005:**
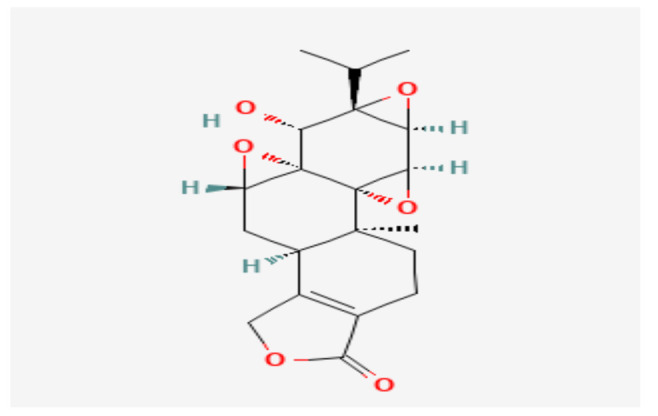
Chemical Structure of Triptolide (PubChem 12966367).

**Figure 6 nutrients-14-00619-f006:**
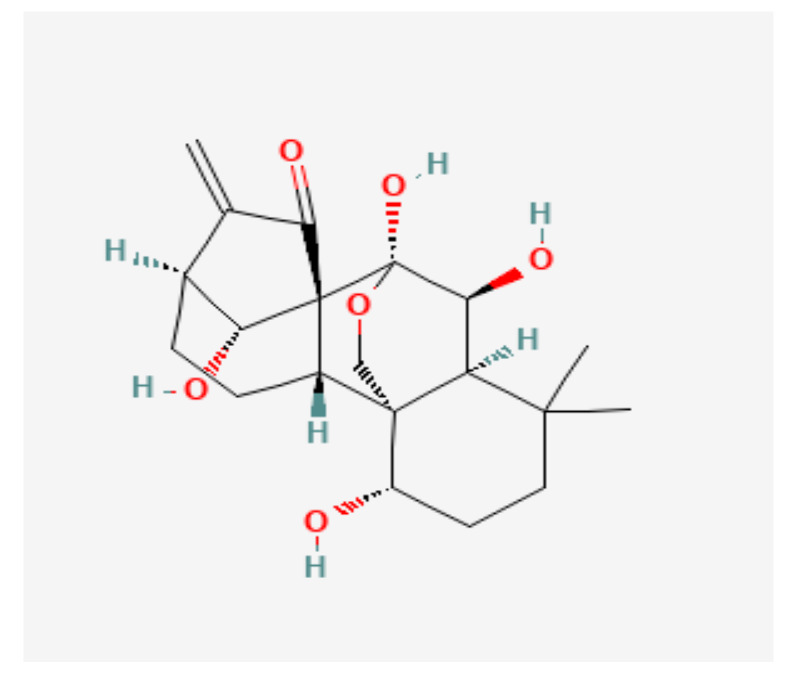
Chemical structure of Oridonin (PubChem 134688823).

**Figure 7 nutrients-14-00619-f007:**
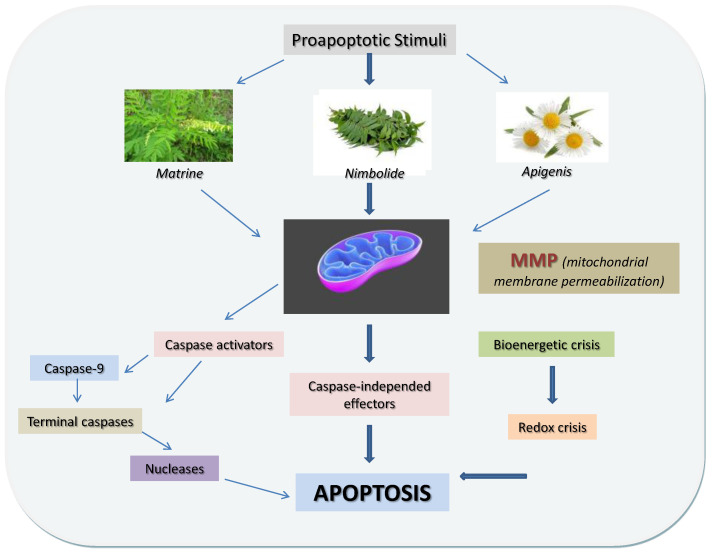
Induction of mitochondrial (“Intrinsic”) pathway of apoptosis induced by various herbal and plant proapoptotic stimuli. The proapoptotic stimuli derived from various plants and herbals (Matrine, Apigenis, Nimbolide, Curcumocin, Lupeol, etc.) initiate this pathway by inducing permeability of MMP (mitochondrial membrane permeabilization). Subsequently, intermembrane space proteins are released into the cytosol, and the mitochondrial transmembrane potential is dissipated, causing, as a result, the induction of bioenergetic and redox crises, leading to the activation of both caspase-dependent and -independent mechanisms, committing the cell to death.

**Figure 8 nutrients-14-00619-f008:**
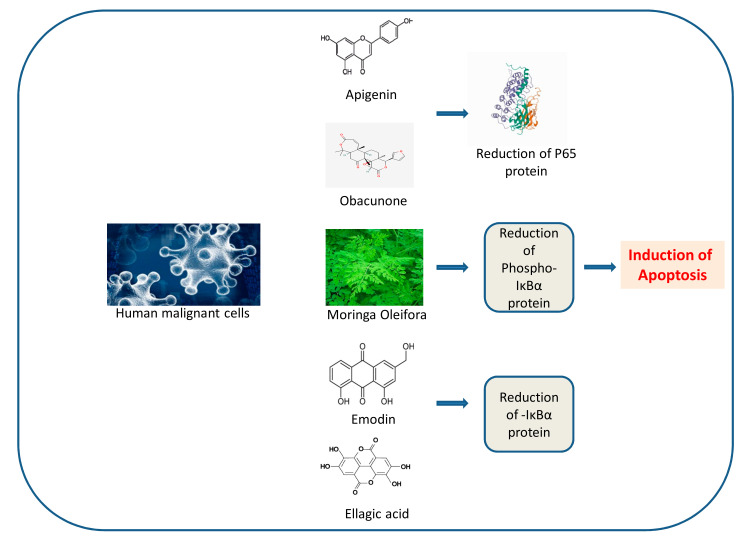
Inhibition of the NF-κB signaling pathway by a number of chemical substances found on herbal and plant extracts. These substances reduce the levels of the expression of a number of key NF-κB family proteins in the cells, including p65, phosphor-IκBα, and IκBα proteins, therefore inducing apoptosis and cell death.

**Table 1 nutrients-14-00619-t001:** Studies combining plants with chemotherapeutic agents (in alphabetical order).

Reference	Herbal	Results	Conclusion
Shimada et al., 2018 [53]	*Asparagus* *extract*	The asparagus extract down-regulated heat shock protein27 in klm1-r cells.	It enhances anticancer effects by combination with gemcitabine in PC.
Thani et al., 2014 [54]	*Chokeberry extract (Aronia melanocarpa*)	Gemcitabine in combination with chokeberry extract was more effective than gemcitabine alone in human pancreatic cell line.	Gemcitabine chemothera-py might be augmented with the chokeberry extract.
Qian et al., 2016 [55]	*Coix seed emulsion*	Coix seed emulsion synergistically sensitized PC cell lines to gemcitabine, both in vitro and in vivo.	Coix seed emulsion sensitized PC cells to gemcitabine therapy.
Pak et al., 2021 [56]	*C5E*	Side population cells and the cell viability of PANC-1 cells were decreased after treatment. mRNA expression levels of sonic hedgehog were significantly down-regulated following the co-treatment.	Combined treatment of gemcitabine and C5E may exhibit synergistic effects in PANC-1 cells.
Wei et al., 2011 [57]	*Emodin*	The combination treatment promoted apoptotic cell death and mitochondrial fragmentation, and reduced phosphorylated-Akt level, NF-κB activation, and Bcl-2/Bax ratio.	Emodin that can either enhance the effects or overcome chemoresistance to gemcitabine.
Rimmon et al., 2013 [58]	*Eskin* (*Aesculus hippocastanum*)	Escin with gemcitabine showed an additive effect, whereas combination with cisplatin had a synergistic cytotoxic effect.	Synergistic effect if combined with cisplatin.
Kim et al., 2018 [59]	*Fisetin*	Combination treatment with fisetin and gemcitabine inhibited proliferation of PC cells, and induced apoptosis. Fisetin sensitized PC cells to gemcitabine-induced cytotoxicity through inhibition of ERK-MYC signaling.	Combination of fisetin and gemcitabine represent a novel therapeutic strategy for PC.
Capistrano et al., 2016 [60]	*Gloriosa ignal L. (glory lily, Colchicaceae*)	Delay in tumour growth for gemcitabine and the combination therapy compared to the control group, and prolongation of the survival.	It has an added value combined with gemcitabine in PC.
Pak et al., 2016 [61]	*Herbal mixture* *extract*	Inhibition of PANC1 cell growth. Suppression of stem cell-like side population cell and migration activity. Suppression of tumor growth in a PANC1-xenograft model.	Possible therapeutic agent for PC and cancer stem cells.
Li et al., 2018 [62]	*Isodon eriocalyx and its bioactive component eriocalyxin b*	Gemcitabine and eriocalyxin B had synergistic anti-proliferative effect. The underlying mechanisms involved included increased activation of the caspase cascade and induction of JNK phosphorylation.	Gemcitabine and eriocalyxin B taken together promoted apoptosis acting synergistically.
Akasaka et al., 2013 [63]	*Monogalactosyl diacylglycerol*	Gemcitabine and monogalactosyl diacylglycerol suppressed growth in PC cell lines. Synergistic effect on inhibition of DNA replicative polymerase inhibitors compared with gemcitabine or monogalactosyl diacylglycerol alone. Pre-addition of monogalactosyl diacylglycerol enhanced cell proliferation suppression by gemcitabine.	Spinach monogalactosyl diacylglycerol could be an effective clinical anticancer chemotherapy in combination with gemcitabine.
Akasaka et al., 2016 [64]	*Monogalactosyl diacylglycerol* *plus radiation*	A dose- and time-dependent cytotoxicity, and reduced cell colonies upon treatment with both monogalactosyl diacylglycerol and radiation as compared to irradiation alone. Higher proportion of apoptosis and DNA damage in pancreatic cancr cells as compared to either one alone.	Enhances the cytotoxicity of radiation in PC cells in vitro and in vivo. Combination with radiation could be effective in PC.
Hagoel et al., 2019 [65]	*Moringa* *olifeira*	Moringa administration combined with radiation therapy significantly inhibited human PC cell survival, induced apoptosis, and reduced metastatic potential. Inhibition of growth of tumors generated by PC cells in nude mice.	Additional inhibitory effect by overcoming the radioresistance of PC cells.
Gong et al., 2017 [66]	*Nexrutine*	Combination treatment of human PC cells with nexrutine and gemcitabine: significant alterations of proteins in the STAT3/NF-κB signaling axis and growth inhibition in a synergistic manner.	The natural extract nexrutine can improve gemcitabine sensitivity.
Hernandez-Unzueta et al., 2019 [67]	*Ocoxin*	It enhances the cytotoxic effect of paclitaxel and gemcitabine, and ameliorates the chemo-resistance in PC cells. It promotes the expression of the altered genes, and decreases tumor development in vivo.	A potential complement to chemotherapeutic agents used in PC
Cheung et al., 2015 [68]	*Oplopanax horridus* (*Devil’s club or devil’s walking stick*)	This extract alone, or in combination with cisplatin, gemcitabine, and paclitaxel, induced toxicity on pancreatic endocrine HP62 and PC. It inhibited proliferation of HP62, PANC-1, and BxPC-3 cells.	It can be used as an adjunct therapy for patients with resistance to conventional chemo-therapeutic agents.
Tai et al., 2014 [69]	*Oplopanax horridus* (*Devil’s club or devil’s walking stick*)	PANC-1 3D spheroids were more resistant to killing by Oplopanax horridus extract, gemcitabine, and paclitaxel compared to 2D cells. It enhanced the antiproliferation activity of cisplatin and gemcitabine. The bioactive compound showed strong antiproliferation activity against PANC-1 2D cells and 3D spheroids.	It enhances the activity of chemotherapeutics against PC cells. 3D spheroid model helps in discovering in vivo bioactive compounds.
Liu et al., 2018 [70]	*Paeonia suffruticosa aqueous extracts*	Alone or in combination with gemcitabine, delayed tumor growth in a xenograft model by stimulating the endoplasmic-reticulum-related proteostasis stress, and inducing autophagy and cell apoptosis.	Potential therapeutic effect in PC in combination with gemcitabine.
Yu et al., 2013 [71]	*Pao Pereira*	Combination with gemcitabine had a synergistic effect in the inhibition of cell growth. In an orthotopic pancreatic xenograft mouse model, gemcitabine did not show inhibition, whereas Pao Pereira suppressed tumor growth. Combined treatment enhanced the tumor inhibitory effect vs. gemcitabine alone.	The extract of Pao Pereira possesses anti-PC abilities, and enhances the effects of gemcitabine both in vitro and in vivo.
Rawat et al., 2020 [72]	*Piperlongumine*	Piperlongumine inhibits cell proliferation, and increases the intracellular reactive oxygen species. P53, P21, BAX, and SMAD4 are up-regulated, whereas BCL2 and signaling are down-regulated.	Piperlongumine with paclitaxel has a synergistic effect.
Jiang et al., 2016 [73]	*Resveratrol*	Resveratrol suppressed proliferation, and induced apoptosis in PC cells. YES-activated protein silencing by resveratrol enhanced the sensitivity of gemcitabine in PC cells.	YES-activated protein is a promising target for sensitizing cancer cells to chemotherapy.
Yu et al., 2014 [74]	*Rauwolfia vomitoria*	*Rauwolfia vomitoria* induced apoptosis in PC models. Combined administration of *Rauwolfia vomitoria* and gemcitabine had a synergistic effect in inhibiting cell growth. *Rauwolfia vomitoria* suppressed tumor growth and metastatic potential in an orthotopic PC mouse model.	The combination reduces tumor burden and metastatic potential in gemcitabine non-responsive tumor.
Mu et al., 2015 [75]	*Thymoquinone*	Pretreatment with thymoquinone following gemcitabine increased the cancer cell apoptosis, and inhibited tumor growth. The combination induced down-regulation of antiapoptotic and up-regulation of proapoptotic molecules.	Thymoquinone pretreatment can enhance the anti-cancer activity of gemcitabine.
Yang et al., 2011 [76]	*Triptolide*	Combined therapy of triptolide and hydroxycamptothecin on PC cell line was superior to that of triptolide or hydroxycamptothecin alone. Activation of caspase-9/caspase-3, and inhibition of NF-κB signaling pathway, were responsible for the synergistic cytotoxic effect.	Combined triptolide and hydroxycamptothecin therapy in patients with PC should be tested.

**Table 2 nutrients-14-00619-t002:** Names, country of origin, and mechanisms of anticancer effects of plant extracts investigated against pancreatic cancer cell lines.

No.	Plant Name	Country of Origin	Mechanism of Action	Reference
1	*Achyranthes aspera*	India	Suppression of the transcription of metalloproteases, and angiogenic factors.	[89]
2	*Alpinia officinarum*	Southeast Asia	Suppression of cell proliferation, and induction of cell cycle arrest.	[90]
3	*Amoora rohituka*	Bangladesh	Induction of apoptosis in PC HPAF-II cells, inhibition of K-ras activity, and suppression of cell proliferation.	[91]
4	*Ancistrocladaceae*	Africa/ Asia	Cytotoxicity against human PC cells under nutrition-deprived conditions. Inhibition of colony formation of PC cells.	[92]
5	*Apigenin*	Global distribution	Induction of the death of pancreatic cell, arrest of the cell proliferation, and induction of apoptosis through mitochondrial pathway.	[93,94]
6	*Asteracea and Lamiaceae*	Global distribution	*Asteraceae* extracts induced cytotoxicity, and inhibited cell transformation.	[95]
7	*Bitter melon juice*	Global distribution	Activation of caspases, decreased signaling and X-linked inhibitor of apoptosis protein levels. Activation of adenosine monophosphate-activated protein kinase.	[96]
8	*BRM270*	Global distribution	Induction of apoptosis in CD44+ cells, inhibition of metastasis traits in CD44^+^ PDAC.	[97,98]
9	*Boesenbergia rotunda*	China/South-East Asia	Cytotoxic action against human PC cells under nutrition-deprived conditions.	[99]
10	*Boswellia sacra gum resins*	China	Reduction of the viability, and increased death after treatment with fractions III and IV of human PC cells. Anti-proliferative and pro-apoptotic activities in the heterotopic xenograft mouse model.	[100]
11	*Brucea javanica*	Sri Lanka, India, China, Australia	Accentuation of the expression of caspase 9 and 3 in Capan-2 cells. Induction of apoptosis in Capan-2 cells through mitochondrial pathway.	[101,102]
12	*Cannabinoids*	Global distribution	Antiproliferative and proapoptotic effects in vitro mediated through various pathways.	[103]
13	*Citrus unshiu Peel*	Japan	Inhibition of growth of PC cells through induction of caspase-3 cleavage. It blocked the migration of the cancer cells through activation of intracellular signaling pathways.	[104]
14	*Cloves* (*Syzygium aromaticum*)	Indonesia	Inhibition of tumor growth in HT-29 xenograft mice model through induction of cell autophagy.	[105]
15	*Cocoa polyphenol*	Global distribution	Decreased the NF-κB transcriptional activity of premalignant and malignant Kras-activated pancreatic ductal epithelial cells.	[106]
16	*Coix ignaling-jobi seed emulsion*	China	Inhibition of NFkB signaling pathway, and inhibition of protein kinase C activity.	[107]
17	*Crocus sativus*	Mediterranean, Asia, Iran	Induction of apoptosis and cell cycle arrest, decreased cell viability.	[108]
18	*Cryptotanshinone*	China	Inhibition of proliferation, and induction of cell apoptosis and cycle arrest in PC cells. Up-regulation of caspase-3 and -9, and poly ADP ribose polymerase, and down-regulation of c-myc, ignaling, and cyclin D1.	[109,110]
19	*CucurbitacinE*	China	Inhibition of STAT3 phosphorylation, and up-regulation of p53 expression.	[111]
20	*Cucurmosin*	America	Induction of apoptosis, inhibition of cell growth, and inhibition of P13K/Akt/mTOR signaling pathway.	[112]
21	*Dandelion root extract*	China and America	Induction of selective apoptosis, as well as collapse of the mitochondrial membrane potential, leading to prodeath autophagy.	[113]
22	*Degalactoti-gonin* *Solanum nigrum*	Asia, America, Australia, South Africa	Inhibition of EGF-induced proliferation, and migration and down-regulation of cuclin D1.	[114]
23	*Diterpene quinones*	Global distribution	KIS37 (cryptotanshinone): Inhibition of KRAS-activated PC cell lines, suppression of KRAS protein, and phosphorylation of Rb and cyclin D1, and PC cell growth in xenografts.	[115]
24	*Elemene*	Global distribution	Up-regulation of tumor expression of P53, and down-regulation of Bcl-2 expression.	[116]
25	*Ellagic acid*	Global distribution	Inhibition of angiogenesis and metastasis in tumor tissues, NF-κB pathway, and COX-2; up-regulation of E-cadherin; and down-regulation of Vimentin.	[117,118]
26	*Emodin Rheum palmatum L*	China	Down-regulation of NF-κB DNA-binding activity, and up-regulation of cleaved caspase-3.	[119]
27	*Eryngium billardieri*	Global distribution	Overexpression of Bax, and underexpression of cyclin D1 on PANC-1 cancer cell lines	[120,121]
28	*Eucalyptus*	Australia	Induction of caspase 3/7-mediated apoptosis	[122,123,124]
29	*Ferula Hezarlaleh-zarica*	Iran	Anti-proliferative action on PANC-1 cells.	[125]
30	*Gallic acid*	Global distribution	Down-regulation of Bcl-2, depolarization of mitochondrial membrane. Reduction of the formation of reactive oxygen species	[126]
31	*Garlic*	Global distribution	Regulation of the JNK protein levels. Inhibition of all PC cell lines proliferation	[127,128]
32	*Gedunin*	India	Induced of anti-metastatic effect through inhibition of sonic hedgehog signaling	[129]
33	*Ginger Extract*	Asia, Africa, America	Induction of ROS-mediated autosis.	[130]
34	Ginkgolic acid	Asia	Down-regulation of the expression of enzymes involved in lipogenesis, and activation of protein kinase signaling.	[131]
35	*Grape proan-thocyanidin*	Global distribution	Reduction of antiapoptotic proteins, and increased expression of Bax.	[132]
*36*	*Graviola*	Tropical countries	Inhibition of multiple signaling pathways regulating metabolism, survival, and metastatic potential of PC cells.	[133]
37	*Green tea extract*	Global distribution	Inhibition of molecular chaperones heat-shock protein 90, and heat-shock protein 27, and inhibition of p53 and Akt.	[134]
38	*Helicteres hirsuta Lour*	Vietnam Cambodia, Indonesia, Thailand	In vitro activity against various PC cell lines.	[135]
39	*Inula helenium*	Eurasia	Inhibition of the phosphorylation of the signal transducer, and activator of transcription (stat)3/akt pathway.	[136]
40	*Lonicera japonica*	China, Japan	Inhibition of BxPC-3 and PANC-1 cell growth.	[137]
41	*Lupeol*	America, Japan, China, Africa	Induction of apoptosis and cell cycle arrest.	[138]
42	*Mangifera indica*	India, Brazil, Africa	Potent cytotoxicity against human PC cells under nutrition-deprived condition.	[139]
43	*Mexican lime*	Mexico	Increased expression of Bax, Bcl-2, casapase-3, and p53, and inhibition of proliferation.	[140]
44	*Moringa Oleifera*	India	Inhibition of NF-κB signaling pathway, and increase of the efficacy of chemotherapy with cisplatin in human PCcells.	[141]
45	*Matrine*	China	Inhibition of cell viability by down-regulation of the expression of PCNA, induction of apoptosis, and increase of activation of caspases-8, -3, -9.	[142]
46	*Naringenin and Hesperetin*	Japan, Spain, China, Korea, S. Africa, America	Inhibition of the phosphorylation of focal adhesion kinase and p38 signaling pathway.	[143]
47	*Nerium oleander*	China, Russia	Potent antitumor activity, through down-regulation of PI3k/Akt and mTOR pathways.	[144]
48	*Nimbolide*	India	Inhibition of proliferation and metastasis via mitochondrial-mediated apoptotic cell death.	[145]
49	*Obacunone*	Global distribution	Induction of apoptosis (activation of caspase-9 and -3, up-regulation of p53, and down-regulation of Bcl2 and NFκB and Cox-2).	[146]
50	*Ocimum sanctum*	India	Up-regulation of genes inhibiting metastasis and inducing apoptosis, and down-regulation of genes promoting survival	[147]
51	*Oleuropein*	Mediterra-nean, China, Asia	Decrease of the viability of the PC cells.	[148]
52	*Olive Biophenols*	Global distribution	Induction of apoptosis of MIA PaCa-2 cells.	[149]
53	*Oridonin*	China	Induction of apoptosis, and inhibition of BxPC-3 cells through regulation of the expression of miRNAs.	[150,151]
54	*Paeonia lactiflora*	Asia, China, Siberia	Inhibition of PC growth by up-regulation of HTRA3.	[152]
55	*Palm oil phenolics*	Tropics	Induction of apoptosis associated with decrease in survivin and Bcl-XL expression.	[153]
56	*Paramignya trimera* and *Phyllanthus amarus*	India	Strong cytotoxic capacity.	[154,155,156]
57	*Plumbagin*	Organic compound	Induction of apoptosis in PC cells through the mitochondria-related pathway	[157]
58	*Pomegranate extract*	India, America	Inhibitory effect through yet unidentified phytochemicals.	[158]
59	*Pulsatilla koreana*	Korea	Increased caspase-3, and decreased of Bcl-2 expression. Decreased expression of major factors of angiogenesis.	[159]
60	*Quercetin*	Global distribution	Sensitized resistant cells to daunorubicin.	[160,161]
61	*Radix Scutellariae*	Asian countries	Induction of apoptosis and autophagy in PC cell lines through PI3K/Akt/mTOR signaling pathway.	[162,163]
62	*Rhazya stricta*	South Asia, Middle East	Reduction in cell viability with dose-dependent manner, and decrease in mRNA expression in PANC-1 and AsPC-1 PC cells.	[164]
63	*Salvia chinensis*	China	Potent cytotoxicity in the MiapaCa-2 human PC cells.	[165]
64	*Sedum sarmentosum Bunge*	China	Increased cellular apoptosis. Induction of p53 expression, and inhibition of epithelial-mesenchymal transition. Down-regulation of the proliferation-related hedgehog signaling pathway.	[166]
65	*Sugiol*	Taiwan	Induction of apoptosis, and up-regulation of the expression of Bax, with down-regulation of Bcl-2 expression.	[167]
66	*TEOA*	China	Inhibition of the proliferation and migration of PC cells, and induction of autophagic cell death in PC cells.	[168]
67	*Toosendanin*	China	It inhibits PC cell growth by blocking Akt/mTOR signaling pathway.	[169,170]
68	*Tripterigium wilfordii*	China	Triptolide-plasminogen activator urokinase could represent a novel target for patients with PC.	[171]
69	*Valtrate*	China	Increased expression of Bax; suppression of Bcl-2, c-Myc, and Cyclin B1; and inhibition of the transcriptional activity of Stat3.	[172]
70	*Xanthohumol*	Europe, Asia, South America	Inhibition of phosphorylation of signal transducer, activation of the transcription 3, and expression of its downstream targeted genes.	[173]
71	*Xao tam phan*	Vietnam	Antioxidant and anti-proliferative activities.	[174]
72	*Xylaria psidii*	It grows on dead wood	Cell cycle arrest and loss of mitochondrial membrane potential.	[175]
73	*Wikstroemia indica*	China, Viet-nam, India, Philippines	Cytotoxicity in the nutrient-deprived medium.	[176]
74	*Ziziphus nummularia*	Saudi Arabia	Inhibition of angiogenesis, reduction of VEGF and nitric oxide levels. Down-regulation of ERK1/2 and NF-κB signaling pathways.	[177]

## Data Availability

No applicable.

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
