# Peer review of "Herbals and Plants in the Treatment of Pancreatic Cancer: A Systematic Review of Experimental and Clinical Studies"

_nutrients, 2022, doi:10.3390/nu14030619_

Round 1

Reviewer 1 Report

Dear Authors, I have read with great interest and curiosity your paper. I think you  provided a very comprehensive view of the potential role of herbals and plants in the treatment of pancreatic cancer, with an extensive literature research effort. Tables are clear and well designed. I have appreciated that you expressed your opinion clearly along the text. However, I believe that due to the limited number of clinical trials assessing the role of herbals and plants and especially their interaction with conventional chemotherapy and polypharmacy in pancreatic cancer patients, you should underline a little bit more that caution is required and more clinical studies are needed before affirming that herbals and plants can be effective strategies in pancreatic cancer and that alternative medicine should enter medical education. Indeed, most studies have been performed in the preclinical setting and at the present time there is insufficient data to draw conclusions on the clinical management of pancreatic cancer patients. Surely, some results are interesting and deserve further research but in my opinion some affirmations are too daring and should be changed with more cautious statements to be appropriate for publication (e.g. lines 63-65: Based on accumulating clinical and experimental data, it seems that the traditional Chinese medicine, as well as many plants growing in other parts of the world, could represent a positive answer to the poor results obtained from the use of the conventional chemotherapy; lines 76-77: We propose that Medical Schools should include alternative medicine and the role of herbals and plants in their regular educational programs; lines 302-304 They found that Parvifloron D has selective cytotoxicity to PC lines possible being an efficient alternative treatment against PC; lines 483-485: The combination treatment of monogalactosyl diacylglycerol and radiation could be an effective strategy for the treatment of PC; line 1279: Leaves of Ocimum sanctum could be used against PC; lines 1667-1669: There is a need for more essential representation of alternative medicine in under- and postgraduate medical education).

other comments: 

  • please revise English and various statements in the manuscript: some words are incomplete and some sentences disaggregated
  • please revise abbreviation use through the text
  • lines 158-161 Werthmann et al also also described a case of a patient 158 with advanced PC with R1-resection, who developed liver metastasis. An overall survival of 63 months and a relapse-free survival of 39 months under Viscum album therapy was achieved:  please be more precise and specify that this patient received also concomitant FOLFIRINOX and then added Viscum album administration, as explained by the Authors 
  • lines 643-658: Patients with pancreatic cancer and an ECOG performance status of 0 to 2 received PHY906 and capecitabine (...) Role of IL-6 in tumor progression and tumor cachexia needs to be investigated with respect to its relation to pathophysiology of pancreatic cancer and development of anti-IL-6 therapeutics: These statements are included in the paragraph 3.5. EXPERIMENTAL STUDIES USING PLANTS AS UNIQUE AGENTS AGAINST 642 PANCREATIC CANCER CELL LINES AND XENOGRAFTS, but the topic seems not related. Please check and correctly reallocate this study in your manuscript.

Author Response

COVER LETTER

Dear Editor, Dear Reviewer,

We wish to re-submit our paper entitled “Herbals and plants in the treatment of pancreatic cancer: A systematic review of experimental and clinical studies modified according to the suggestions of the reviewers.

The authors would like to express their thanks to the reviewers for critical assessment of our work.

We have acted upon the recommendations of the reviewers which have resulted in an enhancement of our manuscript's quality.

All modifications incorporated in the manuscript are highlighted in red and described herein indicating the exact location of each change. A “point-by-point” response to the reviewer’s comments is outlined below.

On behalf of my co-authors, I reiterate my thanks to the reviewer for valuable suggestions and constructive input to improve this manuscript's quality.

Reviewer 1

Dear Authors,

I have read with great interest and curiosity your paper. I think you provided a very comprehensive view of the potential role of herbals and plants in the treatment of pancreatic cancer, with an extensive literature research effort. Tables are clear and well designed. I have appreciated that you expressed your opinion clearly along the text.

We sincerely thank you for your kind comments.

However, I believe that due to the limited number of clinical trials assessing the role of herbals and plants and especially their interaction with conventional chemotherapy and polypharmacy in pancreatic cancer patients, you should underline a little bit more that caution is required and more clinical studies are needed before affirming that herbals and plants can be effective strategies in pancreatic cancer and that alternative medicine should enter medical education. Indeed, most studies have been performed in the preclinical setting and at the present time there is insufficient data to draw conclusions on the clinical management of pancreatic cancer patients.

Since we generally agree with your statement we included your remarks in the text as they were.

Surely, some results are interesting and deserve further research but in my opinion some affirmations are too daring and should be changed with more cautious statements to be appropriate for publication (e.g. lines 63-65: Based on accumulating clinical and experimental data, it seems that the traditional Chinese medicine, as well as many plants growing in other parts of the world, could represent a positive answer to the poor results obtained from the use of the conventional chemotherapy; lines 76-77: We propose that Medical Schools should include alternative medicine and the role of herbals and plants in their regular educational programs; lines 302-304 They found that Parvifloron D has selective cytotoxicity to PC lines possible being an efficient alternative treatment against PC; lines 483-485: The combination treatment of monogalactosyl diacylglycerol and radiation could be an effective strategy for the treatment of PC; line 1279: Leaves of Ocimum sanctum could be used against PC; lines 1667-1669: There is a need for more essential representation of alternative medicine in under- and postgraduate medical education).

Following your suggestion the relevant proposals were significantly modified or even deleted.

Other comments: 

Please revise English and various statements in the manuscript: some words are incomplete and some sentences disaggregated.

The whole paper was revised by a professional translator

Please revise abbreviation use through the text

The abbreviation use was revised

Lines 158-161 Werthmann et al also described a case of a patient 158 with advanced PC with R1-resection, who developed liver metastasis. An overall survival of 63 months and a relapse-free survival of 39 months under Viscum album therapy was achieved:  please be more precise and specify that this patient received also concomitant FOLFIRINOX and then added Viscum album administration, as explained by the Authors.

The exact kind of therapy of the patient was mentioned in the text

Lines 643-658: Patients with pancreatic cancer and an ECOG performance status of 0 to 2 received PHY906 and capecitabine (...) Role of IL-6 in tumor progression and tumor cachexia needs to be investigated with respect to its relation to pathophysiology of pancreatic cancer and development of anti-IL-6 therapeutics: These statements are included in the paragraph 3.5. EXPERIMENTAL STUDIES USING PLANTS AS UNIQUE AGENTS AGAINST 642 PANCREATIC CANCER CELL LINES AND XENOGRAFTS, but the topic seems not related. Please check and correctly reallocate this study in your manuscript.

The study is included in the paragraph 3.1 CLINICAL STUDIES USING PLANTS IN THE TREATMENT OF PANCREATIC CANCER (3.1.1. PHY906)

Reviewer 2 Report

Although the study looks interesting, there are some issues with the findings

  • I would strongly recommend to include at least 2 or more schematic diagram describing mechanistic pathways of phytochemicals.
  • Author has mentioned several compounds. However, they missed the chemical structure of compounds. Therefore,  I recommend to draw the chemical structures of at least those which are not commonly known.
  • Author must have to mention the limitations of their survey or review in the discussion section.
  • Author has to discuss future perspective of those compounds that are currently under trial or finished clinical trial.

Author Response

Dear Editor, Dear Reviewer,

We wish to re-submit our paper entitled “Herbals and plants in the treatment of pancreatic cancer: A systematic review of experimental and clinical studies modified according to the suggestions of the reviewers.

The authors would like to express their thanks to the reviewers for critical assessment of our work.

We have acted upon the recommendations of the reviewers which have resulted in an enhancement of our manuscript's quality.

All modifications incorporated in the manuscript are highlighted in red and described herein indicating the exact location of each change. A “point-by-point” response to the reviewer’s comments is outlined below.

On behalf of my co-authors, I reiterate my thanks to the reviewer for valuable suggestions and constructive input to improve this manuscript's quality.

Reviewer 2

I would strongly recommend including at least 2 or more schematic diagram describing mechanistic pathways of phytochemicals.

Two schematic diagrams describing the mechanistic pathways were added.

Author has mentioned several compounds. However, they missed the chemical structure of compounds. Therefore, I recommend drawing the chemical structures of at least those which are not commonly known.

The chemical structure of all compounds was added and described.

Author must have to mention the limitations of their survey or review in the discussion section.

The strangths and limitations of the study were added in the discussion section.

Author has to discuss future perspective of those compounds that are currently under trial or finished clinical trial.

The future perspectives were also mentioned in the discussion section. Unfortunately we were not able to find out ongoing clinical trials.
